# Impedimetric and Plasmonic Sensing of Collagen I Using a Half-Antibody-Supported, Au-Modified, Self-Assembled Monolayer System

**DOI:** 10.3390/bios11070227

**Published:** 2021-07-08

**Authors:** Marcin Gwiazda, Sheetal K. Bhardwaj, Ewa Kijeńska-Gawrońska, Wojciech Swieszkowski, Unni Sivasankaran, Ajeet Kaushik

**Affiliations:** 1Faculty of Materials Science and Engineering, Warsaw University of Technology, 141 Woloska Str., 02-507 Warsaw, Poland; marcin.gwiazda@postgrad.manchester.ac.uk (M.G.); ewa.kijenska@pw.edu.pl (E.K.-G.); wojciech.swieszkowski@pw.edu.pl (W.S.); 2Department of Chemistry, The University of Manchester, Manchester M13 9PL, UK; 3Institute of Animal Reproduction and Food Research, Polish Academy of Sciences, Tuwima 10, 10-748 Olsztyn, Poland; unni.siva.info@gmail.com; 4Van’t Hoff Institute for Molecular Sciences, University of Amsterdam Science Park 904, 1098 XH Amsterdam, The Netherlands; 5Centre for Advanced Materials and Technologies CEZAMAT, Poleczki 19, 02-822 Warsaw, Poland; 6NanoBioTech Laboratory, Department of Natural Sciences, Florida Polytechnic University, Lakeland, FL 33805, USA

**Keywords:** collagen type I, 4,4′-thiobisbenzenethiol, nanogold, electrochemical impedance spectroscopy, surface plasmon resonance, half antibody, medical diagnostic devices

## Abstract

This research presents an electrochemical immunosensor for collagen I detection using a self-assembled monolayer (SAM) of gold nanoparticles (AuNPs) and covalently immobilized half-reduced monoclonal antibody as a receptor; this allowed for the validation of the collagen I concentration through two different independent methods: electrochemically by Electrochemical Impedance Spectroscopy (EIS), and optically by Surface Plasmon Resonance (SPR). The high unique advantage of the proposed sensor is based on the performance of the stable covalent immobilization of the AuNPs and enzymatically reduced half-IgG collagen I antibodies, which ensured their appropriate orientation onto the sensor’s surface, good stability, and sensitivity properties. The detection of collagen type I was performed in a concentration range from 1 to 5 pg/mL. Moreover, SPR was utilized to confirm the immobilization of the monoclonal half-antibodies and sensing of collagen I versus time. Furthermore, EIS experiments revealed a limit of detection (LOD) of 0.38 pg/mL. The selectivity of the performed immunosensor was confirmed by negligible responses for BSA. The performed approach of the immunosensor is a novel, innovative attempt that enables the detection of collagen I with very high sensitivity in the range of pg/mL, which is significantly lower than the commonly used enzyme-linked immunosorbent assay (ELISA).

## 1. Introduction

Currently, significant medical developments require the use of modern technologies to continuously monitor a patient’s state. Accordingly, it is highly demanded to construct and apply wearable sensors or in situ biomarkers, which are capable of detecting targets through real-time sensing [1,2,3]. One such crucial biomarker is collagen. This protein is distributed in different human tissues, such as bone, cartilage, tendons, ligaments and the cornea, and plays an important role in the regeneration of damaged connective tissue in tendons and ligaments [4,5,6]. Moreover, collagen is contained in the composition of the extracellular matrix (ECM) and ensures the high mechanical properties required for connective tissue [7]. Collagen type I triple helix consists of two identical α1(I)-chains and one α2(I)-chain. Thus, it provides relatively good biomechanical properties, including load-bearing, tensile strength and torsional stiffness [4,8]. Collagen fibrils are directly responsible for the mechanism of tendon regeneration [9]. Additionally, the determination of collagen type I protein has an important function in the case of tendon inflammation [8,9,10]. Tendons and ligaments exhibit high tensile strength [11] because they are mostly composed of connective tissue, proteoglycans, elastin and collagen type I and III fibrils with spindle-shaped tenocytes. [12,13] Nevertheless, this structure contributes to the low vascularization of those tissues and reduce their capability for efficient regeneration. [14] To improve and accelerate the regeneration mechanism, it is demanded to initially very precisely evaluate the presence of collagen type I [15], which indicates the occurrence of a mechanism for the synthesis of the collagenous fibrils and thus starting the self-healing process. [10] This directly determines the appropriate selection of applied therapies [16] or invasive surgical reconstruction. [15] The currently performed techniques are based on a biopsy of the damaged tendon or ligament tissue and usually allow to determine the collagen content with the maximum accuracy of ng/mL. [17] Accordingly, the application of in situ, direct biomarkers to evaluate the content of the collagen type I [18] with a highly sensitive level in the picomolar range can significantly improve the diagnosis of the occurrence of the potential healing process and allow for the appropriate treatment sooner using either percutaneous injection of collagen and hyaluronic acid or the implementation of invasive surgery [16]. The presence of even a very small amount of collagen type I, in the picogram range, can induce and accelerate the selection of the appropriate treatment earlier, to apply, for example, different doses of hyaluronic acid or collagen type I/III injection, aiming to avoid the implementation of highly invasive surgery. [10,15,16] To increase the retreatment process, it is significantly relevant to indicate the concentration of collagen type I that directly enhances the regeneration of damaged tendon tissue. For this reason, it is required to implement sensitive systems suitable for detecting collagen type I with an accurate level of selectivity. 

The development of chemical, physical, histochemical and immunochemical methods allowed for the detection of collagen in the micromolar concentration range at the end of the last century [19,20]. Afterward, the application of the enzyme-linked immunosorbent assay (ELISA) based on hydroxyproline allowed for the verification of collagen proteins in the range of µg/mL [21,22]. Currently, electrochemical sensing platforms [23,24] that incorporate specific antibodies responsible for selective antigen detection are at the centre of research interest. These kinds of sensors are highly effective tools to recognise the target analyte, with significant sensitivity and selectivity [1,2,25,26]. However, a major challenge is related to the successful immobilization of biomolecules (enzyme/antibodies/DNA) on the surface of the applied transducer [25,26,27,28,29,30]. For this purpose, it is fundamental to modify the surface of the working electrode to enhance the connection with the antibody receptor. Common methods are based on ‘lock and key’ approaches, such as G and A proteins, which enable immobilization of the receptor antibody on the surface of the transducer [31,32,33,34,35]. The application of G and A proteins is highly efficient for non-covalently binding antibodies, also providing to orientate them on-tail [36]. Hence, those intermediate proteins have five and two specific domains, which allow them to appropriately terminate with the crystallizable region (Fc) and also support the orientation of the on-tail antibodies [37]; this ensures obtaining a uniform arrangement of the receptor of their antigen-binding fragment (Fab) against the complementary antigen, consequently increasing their sensitivity properties. [38] Nevertheless, the high molecular weights of the intermediate proteins and their insulating electric properties [39] between the redox marker and transducer substrate have a negative influence on the electrochemical sensor’s performance, revealing higher electron transfer resistance through Electrochemical Impedance Spectroscopy (EIS).

Alternative solutions are sensors based on redox-active platforms, which are responsible for the immobilization of antibodies as well as for the transduction of analytical signals. Their main advantage is the ability to work without applying redox markers in the sample [40,41,42,43]. Furthermore, there have been different attempts to allow for biosensing that excluded additional redox species [44], such as label-free, impedance-derived redox capacitance for Flavivirus dengue detection [45]. Another appropriate approach is related to the immobilization of antibodies directly on the surface of the transducer through metal nanoparticles [46,47,48,49,50]. These surface forms have high electrical conductivity and compatibility towards protein receptor molecules [51,52,53,54]. In addition, the preformed surface with its noble metal nanoparticles [55] is a capable environment for maintaining the physiological activity of immobilized proteins. Metal nanoparticle monolayers are suitable for the immobilization of whole molecules of antibodies via electrostatic interactions [52,53], as well as for covalent immobilization of the Fab part of antibodies via Au–S covalent bonds [51,54]. The main benefit of using the F_ab_ parts of antibodies is the immobilization of the stable sensing elements, ensuring their appropriate orientation [52,55,56]. 

Moreover, the application of the 4,4′-thiobisbenzenethiol (TBBT) AuNPs SAM [56,57] offers a significant advantage compared to the commonly used 1,6-hexanedithiol (HDT) AuNPs SAM because of its relatively lower electron transfer resistivity [51,58,59,60]. Furthermore, the efficiency of the sensitivity properties could be related to the physical features of the used nanomaterial, which directly depends on the size of the nanoparticles and their electrical conductivity. Application of the SAM gold nanoparticles reveals high electrical conductivity and low resistivity (2.44 × 10^−6^ Ω∙cm) [61,62], which consequently improved the analytical parameters of the immunosensors and sensitivity properties. Application of system-based, orientated, half-reduced antibodies allows obtaining a highly specific immunosensor to the target protein. The recognition of the analyte contributes to the decreasing of the accessibility of the redox-active marker registered through the Electrochemical Impedance Spectroscopy (EIS). Analogically, the highly sensitive orientation of a half-antibody contributes to the efficiency of the interaction with complementary antigens, inducing the mechanism of plasmonic resonance and allowing to detect the shift SPR angle. Consequently, it is not required to use any additional amplification mechanism to verify the changes in the optical or electrochemical signals towards various concentrations of the analyte.

In this work, we construct a biosensor composed of 4,4′-thiobisbenzenethiol (TBBT) SAM, enabled to covalently bound gold nanoparticles (AuNPs). It has been applied for immobilization of half-antibody fragments via metal nanoparticles using disulphide-bridge covalent bonds. The half IgG was derived by the process of the enzymatic digestion using tris(2-carboxyethyl) phosphine hydrochloride (TECP) [63]. In this approach, the further separation process of the reaction mixture was not necessary because the other fractions of the half-Fc and Fab with half-Fab antibody fragments contribute to the mechanism of efficiently blocking empty spaces onto the receptor layer. Additionally, the mechanism of the covalently bonded, reduced IgG half-antibody is stable over the wide range of pH. This is an important advantage in comparison to the electrostatic immobilization, where the various pH ranges have an impact on the isoelectric point of the whole IgG antibody, and it determines the specific value of pH that allows for the stable, controlled attachment of the receptor onto the transducer substrate.

The main aim of the present research is to fabricate an immunosensor that can verify the collagen I concentration under conditions of a low limit of detection, and to be compatible for detection using two different electrochemical and optical methods. This universal system can allow to quickly verify the collagen I content, with the miniaturized portable electrochemical device based on an evaluation of the accessibility of the redox marker to the sensor surface through the EIS measurement of the electron transfer resistance corresponding to the appropriate value of the analyte concentration. Implementation of Electrochemical Spectroscopy Impedance as a quantifying method enables to record very precisely the electrochemical signal of the electron transfer resistance with high sensitivity. [64] The adapted parameters of EIS, such as bias potential and frequency, ensure not having any negative impact on the stability of the receptor layer, in comparison to cyclic voltammetry (CV), which requires a wide range of applied potential. [65] Impedance spectroscopy plays an important role to evaluate the electrochemical condition, stability of the sensor electrode and to detect the rate of charge transfer, absorption of the proteins, ion exchange and interaction between the antibody–antigen recognition. [66] All of those factors have as a fundamental aspect the utilization of the EIS technique to obtain a highly sensitive and specific biosensor, where the determined values of the electron transfer resistance are correlated with the accessibility of the used redox marker to the electrode interface, allowing to very accurately verify the concentration of the analyte. [67] Subsequently, Electrochemical Impedance Spectroscopy was selected as one of the methods used in the collagen I-sensing platform. Moreover, the same biosensing system using an independent optical technique was applied, where the interaction between the specific half-reduced antibody and antigen is generated through excitation of the plasmonic effect shift of the recorded SPR angle. This duality of the verification of the collagen I content supports applying it as a multipurpose system and gives the opportunity to translate it into other biosensing platforms. Furthermore, this extraordinary approach consisted of the application of stable covalent immobilization of AuNPs through the thiol groups and half-reduced monoclonal antibodies. Additionally, the Au layer of the AuNPs ensured an increased surface area of the working electrode and consequently immobilized more half-reduced antibody receptors in comparison to the plain surface [68,69,70]. AuNPs have also been used for the enhancement of the SPR signal response, which leads to enhance the sensitivity and specificity for biomolecule detection. AuNPs favour biosensing amplification due to their plasmonic properties and large dielectric constant. The conducted experiments have value for the construction of novel immunosensors based on the half-IgG reduced antibody for collagen I detection, to obtain high stability, sensitivity and selectivity properties.

## 2. Materials and Methods

### 2.1. Reagents and Materials

Gold colloidal solution (AuNPs, 0.01% concentration of nanoparticles with a 20 nm diameter), monoclonal antibody Anti-Collagen Type I (IgG1 isotype) produced in mouse, bovine collagen solution Type I, tris(2-carboxyethyl) phosphine hydrochloride (TECP), ferro- and ferricyanides and bovine serum albumin (BSA) was procured from Sigma-Aldrich (Poland). Methanol, potassium hydroxide, ethanol and sulfuric acid were supplied by POCh (Poland). Alumina polishing suspensions of 0.30 μm and 0.05 μm were obtained from Buehler (USA). Milli-Q water with a resistivity of 18.2 MΩ·cm^−1^ was purchased from Millipore, Germany. The 4,4′-thiobisbenzenethiol (TBBT) compound was received from Leuven University [51]. All the utilized chemical reagents and solvents were used without any further specific purification because they were of analytical quality. All the experiments of this research were conducted at room temperature.

### 2.2. Preparation of the Half-Antibody Fragment

In order to obtain a half-antibody fragment, the procedure described by Sharma et al. [63] (Figure 1) was applied. Initially, a 5 mM TECP solution was prepared in 10 mM PBS buffer. Then, 2 µL of 5 mM TECP was mixed with 200 µL of 10 µg/mL monoclonal collagen type I antibody in PBS buffer. Afterward, the solutions of TECP and monoclonal collagen type I were mixed at room temperature for 1 h. TECP (‘bond breaker’) reduced the disulphide bridges from both the F_c_ and F_ab_ parts of the antibody. Consequently, the suspension contained a mixture of half-antibody chains of the F_c_ and F_ab_ regions. The separation of the half-IgG and F_ab_ parts from the reaction mixture was not necessary [63].

### 2.3. Fabrication of the Immunosensor to Estimate Collagen I

The electrodes were polished using alumina slurries (0.30 μm) for 15 min followed by gentle washing with methanol and Milli-Q water. Then, this procedure was conducted again, but using 0.05 μm of alumina slurries, to obtain a smooth gold surface following the rinsing of the electrodes with methanol and Milli-Q water. Afterwards, electrochemical cleaning of the Au electrodes was performed by cyclic voltammetry (CV) using an AutoLab potentiostat/galvanostat. CV cycles (*n* = 100) were performed by immersing the working electrode in a 0.5 M KOH solution with an applied potential from 0.4 V to −1.2 V using a 0.1 V/s scan rate. Thereafter, the Au electrode was again electrochemically cleaned using a 0.5 M H_2_SO_4_ solution. Eventually, the Au surface of the working electrode was pre-treated by the activation through the application of 10 CV cycles in a 0.5 M KOH solution. This stage ensured removing any residual impurities absorbed on the gold electrode surface. Thereafter, the working electrode was cleaned by the rising of the Milli-Q water and ethanol. Subsequently, the Au electrode was immersed in a solution of 1.0 mM 4,4′-thiobisbenzenethiol (TBBT) in ethanol for 0.5 h. Then, the electrode was rinsed with ethanol and Milli-Q water. 

Once the SAM layer of TBBT was created on the Au/TBBT substrate, the electrode was flipped to spot the top with 10 µL droplets of the Au colloid solution of AuNPs (Au/TBBT/AuNPs) for 2 h. In the following step, 10 µL droplets of the 10 µg/mL half-antibody fragment solution were used, immobilized for 2 h directly on the surface of Au/TBBT/AuNPs, to obtain the Au/TBBT/AuNPs/half-IgG electrode, and which was further rinsed with PBS buffer. In total, 10 µL droplets of a 1% solution of bovine serum albumin (BSA), dissolved in 0.1 M PBS, pH 7.4, were placed on each electrode (Au/TBBT/AuNPs/half-IgG/BSA) for 0.5 h to block unspecific binding. The overall scheme of the electrode modification steps is represented in Figure 2. Eventually, the modified electrodes were rinsed using a solution of 0.1 M PBS. 

Once the process of the modification was completed, the electrodes were immersed in 0.1 M PBS, and after that, they were incubated in a refrigerator at +4 °C overnight. Electrochemical measurements (CV and EIS) were carried out after each stage of the modification to confirm the successful fabrication of the sensor. 

### 2.4. Collagen I Detection Using Au/TBBT/AuNPs/half-IgG

After fabrication, the electrodes (*n* = 6) were exposed to various concentrations of collagen I (range of concentration: 1, 2, 3, 4 and 5 pg/mL). For the interaction of antigen (collagen I) with half-IgG, 10 µL drops of collagen I, diluted in 0.1 M PBS buffer, were deposited on the modified electrode surfaces. Then, the electrodes were prevented from air contamination and evaporation of the solutions by covering them with black Eppendorf tubes. The incubation time was 30 min at room temperature. After that, the remaining unbound antigens were removed from the electrode surfaces by rinsing with 1 mL of a 0.1 M PBS buffer (pH 7.4).

### 2.5. Electrochemical Measurements

The applied electrochemical system was based on the AutoLab potentiostat/galvanostat (Eco Chemie, Netherlands) with three combined electrodes. The working electrode was represented by the circular electrode with a 2 mm diameter made from polycrystalline Au, the Ag/AgCl as the reference electrode and the platinum wire as a counter electrode. Electrochemical experiments were performed in the electrolyte composed of 0.1 M PBS (aqueous salts solution with 2.7 mM KCl, 137 mM NaCl, 1.8 mM Na_2_HPO_4_, 10 mM KH_2_PO_4_ and pH 7.4) with the addition of 0.5 mM ferro- and ferricyanides (K_3_[Fe (CN)_6_]/K_4_[Fe (CN)_6_]; (1:1)) as a redox-active probe. The cyclic voltammetry (CV) measurements were recorded in the potential range from 0.6 V to −0.2 V at a 0.1 V/s scan rate. Electrochemical Impedance Spectroscopy (EIS) was conducted to determine the value of the electron transfer resistance (R_et_) in a frequency from 0.01 Hz to 100 kHz at 0.17 V of the bias potential for 10 mV of the ac amplitude. The concentration of collagen I was determined by EIS measurements. EIS spectra were fitted using a specific equivalent circuit supported by AutoLab Metrohm NOVA software to determine the value of the electron transfer resistance (R_i_). The response of the immunosensor toward collagen I is represented as (R_i_ − R_0_)/R_0_ × 100%, where R_0_ is the electron transfer resistance for the electrodes after all the steps of the modifications before the detection of any analyte; and R_i_ is related to the electron transfer resistance of the completely modified electrodes after the application of a particular concentration of the analyte.

### 2.6. Atomic Force Microscopy Analyses

Atomic Force Microscopy (AFM) was performed using a Universal SPM Quesant (Agoura Hills, CA, USA), on mica plates coated with thin films of Au (111) of 200 nm thickness. Before the analysis, the Au surfaces were annealed with a hydrogen flame followed by cleaning in an ozone/UV chamber. The intermittent-contact mode was used with a NSC16 tip (CW2, Si_3_N_4_) as a cantilever. A bare Au surface, TBBT-modified Au surface and AuNPs attached to the TBBT-modified Au surfaces were analysed immediately after preparation. The average roughness (R_a_) and thickness from each modification stage were measured. A significant statistical difference (at *p* < 0.005) was evaluated by one-way ANOVA with a post-hoc Tukey test.

### 2.7. Surface Plasmon Resonance Measurements

Surface Plasmon Resonance (SPR) analyses were performed in an Autolab Springle SPR system (Eco Chemie, Netherlands) coupled with a thermostatic water bath at a wavelength of 670 nm and constant temperature of 25 °C. The sensing surface (Au/TBBT/AuNPs/half-IgG/BSA) was placed inside the cuvette, and the sample solution (collagen I) was injected on the active surface at a flow rate of 20 μL/s. After each measurement, the surface was washed with phosphate buffer by switching the flow back to the buffer. The total volume of each sample injection was kept at 100 μL. All the measurements were repeated three times.

## 3. Results

### 3.1. Characterization of Nano-Enabled SAM Using Atomic Force Microscopy

AFM analyses were conducted to study the surface topography of the modified electrodes. These studies were made on bare and modified (with TBBT SAM and TBBT SAM together with AuNPs) Au substrates. For comparison, a 2 μm × 2 μm area scanning was performed and is represented in Figure 3. The average roughness parameter from five different scans was calculated to study the porosity/topography of the obtained surfaces. Bare Au showed terraces on the surface over the scanned area of 0.1 by 0.1 μm (Appendix A). The summary comparison with the measured values for the average roughness (R_a_) and thickness are presented in Appendix A, with the marked presence of the statistically significant differences. The average roughness parameter for the bare Au surface was found to be 21.3 ± 0.7 nm (Figure 3A). The obtained roughness (19.4 ± 3.6 nm) and thickness (21.1 ± 6.3 nm) after the immobilization of TBBT on the Au surface (Figure 3B) was not significantly changed and it was in the same range in relation to the plain gold surface. After the incorporation of Au nanoparticles, the topography of the TBBT-modified surface again changed, and the attached particles became visible in the corresponding AFM images. The roughness of the modified surfaces changed to 22.6 ± 3.8 nm and the measured thickness significantly increased to 27.6 ± 4.3 nm in comparison to the Au/TBBT sample for AuNPs (Figure 3C), which correspond to similar research performed by Park et al. [71] for a SAM AuNP layer. The value of the average roughness was increased in comparison to the previous stage of the functionalization of the electrode surface using the TBBT compound. The condition of the bare Au surface and occurrence of the scratches provided for the relatively high roughness of the base modification substrate. The diameter of the used gold nanoparticles is commonly determined for a colloid solution by the dynamic light scattering (DLS) technique [72]. Consequently, the appearance of the surrounded ions onto the surface of AuNPs led to obtaining a higher hydrodynamic diameter in the range of approximately 20 nm, although the average dimension of the particles usually has quite a wide distribution [73]. In our study, the performance of the AFM measurements was important to validate the mechanism of the formation of the self-assembled monolayer of AuNPs after modification of the Au substrate by the TBBT compound. The heights of the particles obtained from the analysis of the AFM images (at lower area) confirm the presence of nano-sized materials on the modified surfaces. The size of the particles was achieved from AFM 2D images. The corresponding 2D images of the 3D images in Figure 3C are given in Appendix A. The average length and breadth of the particles for the AuNP-modified Au surface were 411.5 ± 35.6 nm and 168.6 ± 38.7 nm, respectively. These results indicate the formation of a larger aggregated cluster of the combined gold nanoparticles, which are arranged in a highly packed density on the surface of the electrode. 

It was observed that the magnitude of the length and breadth showed the aggregation of particles on the surface, contributing to the increase in surface roughness and surface area of the previously plain smooth Au substrate. Therefore, the higher surface area and roughness of the formed Au self-assembled monolayer had a significant impact by binding more half-antibody receptors on the transducer surface, consequently amplifying the received signal and obtaining a much higher sensitivity towards to collagen type I. 

### 3.2. SPR-Assisted Confirmation of Half-Antibody Fragment Collagen I Immobilization

SPR is a direct, label-free, real-time measurement of binding kinetics and affinity. It is an optical detection method that utilises the conjugation of prisms that permit biomolecular interactions in real time. The interaction between biomolecules is analysed by determining the change in the refractive index in real time. This change in refractive index is obtained from the interaction between the immobilized biomolecule and the analyte. It is the most convenient tool to study the interfacial interaction between the analyte (antigen) and the immobilized biomolecules (antibody) in real time [74,75,76,77]. Therefore, we have applied this technique to confirm the immobilization of half-IgG collagen I antibodies on the transducer surface [78,79,80,81]. Upon deposition of the reaction mixture obtained after antibody digestion using TECP [63], an increase in the SPR angle was observed. Real-time, label-free biomolecular interactions between half-IgG and collagen I were recorded using an Auto lab Springle SPR system (Eco Chemie, Netherlands). A 50 nm-thick, gold-coated glass disc was supplied along with the instrument. It is an open cuvette-based dual channel system, where channel-1 was used to measure the interactions between half-IgG and collagen I and channel-2 was used to monitor the signals due to changes in the refractive index of the buffers, and also acted as a reference. Different reagents, samples and buffers were injected in the desired amounts into two cuvettes (assembled over the gold disc). This SPR technique is used to characterize the binding interactions between half-IgG and collagen I without any labelling requirements. 

The SPR angle increased from 10° to 155° after the immobilization of the antibody on the surface of Au. After the deposition of AuNPs on the Au surface, the angle shifted from 10° to 100°, and later, after immobilization of the half-IgG antibodies (Au/TBBT/AuNPs/half-IgG), the angle shifted to 155°, as shown in Appendix A. After the immobilization of half-IgG, steady-state conditions were obtained, and the surface was washed using PBS buffer to remove any unbounded species. The obtained results validated the successful half-IgG antibody immobilization on the Au/TBBT/AuNPs surface within approximately 1 h and 20 min (Appendix A). The shift in angle at each step is shown in Appendix A and described in the Appendix A. In the next step, the time of interfacial interaction between half-IgG and collagen I was checked using SPR. The golden disc support, modified with TBBT/AuNPs/half-IgG/BSA, was placed into an SPR chamber filled with 100 µL PBS buffer. For the specific interaction, the SPR response of the TBBT/AuNPs/half-IgG/BSA sensing platform at various concentrations of collagen type I was recorded in PBS buffer. While the Au/TBBT/AuNPs/half-IgG/BSA surface was exposed to various concentrations of collagen I, the obtained sensogram revealed three phases for each concentration: (i) baseline (1st phase); (ii) association of collagen I with the Au/TBBT/AuNPs/half-IgG/BSA surface (2nd phase); and (iii) dissociation of collagen I (3rd phase). With the inoculation of 1 pg/mL of collagen I, an increase in angle shift is observed from 10° to 40° during the association phase. The SPR response signal increases consistently upon the exposure of collagen I up to 5 pg/mL; after that, the response signal decreases, as shown in Figure 4A. The decreasing value of the SPR angle shift could be caused by the adherence and interaction of the target collagen protein with the binding part of the antibody on the receptor layer. Accordingly, the detection of the analyte in the higher concertation range was impeded. Figure 4B depicts the calibration curve of the SPR signal attained as a function of the collagen I concentrations and signifies linearity between 1 pg/mL and 5 pg/mL. After the association phase, the residual analyte is discarded by using a flushing buffer to clean the surface. The antigen–antibody interaction was measured by injecting collagen I for 6 to 20–25 min followed by a rinsing period of 10 min with pure running buffer. Each experiment was repeated thrice.

The Au/TBBT/AuNPs/half-IgG/BSA immunosensor shows a linear range between 1 and 5 pg/mL; beyond this, it shows a decrease in the SPR angle shift. This decrease in angle after 5 pg/mL shows that the device cannot detect collagen I beyond 5 pg/mL. In our approach, we used the SPR technique to determine the actual dynamic range of the performed collagen immunosensor and consequently validated it through impedance spectroscopy. Therefore, we have decided to conduct the experiment with a range of collagen concentration from 1 to 5 pg/mL. Based on these results, the concentration range of collagen I from 1 pg/mL to 5 pg/mL was selected for the electrochemical sensing, with a time of 30 min for the half IgG–collagen I interaction. The lowest concentration detected by the SPR technique was 1 pg/mL. The same surface was reused several times for the measurements (Au/TBBT/AuNPs). The fabricated Au/TBBT/AuNPs/half-IgG/BSA immunosensor is highly sensitive. Most of the extant research report on a full-antibody-immobilized immunosensor, which leads to random antibody immobilization on the transducer surface [82,83,84]. However, in the present work, an Au/TBBT/AuNPs/half-IgG/BSA immunosensor, thiol (−SH) group of the antibody is conjugated onto the Au surface. The half-IgG antibody immobilization can precisely bind the F_c_ part of the antibody to accomplish an oriented antibody immobilization, as shown in Figure 2. The oriented intact antibody shows higher sensitivity as compared to full-antibody immobilization. It is reported that the half antibody shows a 3–8 times higher sensitivity than the full antibody [85].

### 3.3. Electrochemical Characterization of the Immunosensor Fabrication

The crucial parts of the immunosensors are the specific antibodies responsible for selective recognition of the antigens, as well as the transducer layers, which are mainly responsible for sensor sensitivity. The focus of this work was to verify the optimized stage of the electrode modification and detection of collagen I. Application of the 4,4′-thiobisbenzenethiol (TBBT) compound contained two SH groups that played an important role in the covalent deposition on the Au substrate, as well as covalent immobilization of the AuNPs.

The parameters characterizing TBBT SAM are superior when compared to 1,6-hexanedithiol SAM, widely applied in immunosensor fabrications [51,58,59,60,62]. It was confirmed that TBBT SAM has a charged transfer resistance of approximately 500 kΩ. This value is three times lower than the charged transfer resistance of 1,6-hexanedithiol dithiol [51,52]. Accordingly, it enables receiving a higher electrical signal, detecting the analyte across a wider concentration range and reduced the potential negative effect of electrode blocking. In addition, the foundation of the TBBT SAM on the Au electrode platform is more reproducible and it amplifies the intensity of the recorded signal, because of higher electrical conductivity in comparison to the 1,6-hexanedithiol dithiol SAM. Therefore, TBBT SAM was applied in the present research.

Another important point of immunosensors is the stable immobilization of specific antibodies to maintain their right orientation and physiological activity at the same time [59,60]. The AuNPs SAM has a minus charge due to the citrate anions used for nanoparticle stabilization, which creates an environment suitable for the electrostatic immobilization of antibodies. For these approaches, the selection of the pH conditions preserves the plus charges on the F_c_ part of the antibodies and allows for the right immobilization of whole antibodies with F_ab_ parts, which are responsible for antigen recognition, exposed in the sample solution [53,54,64].

The AuNPs SAM is also very suitable for covalent immobilization of the F_ab_ parts in which the disulphide groups are incorporated [51,54]. To apply this method, the enzymatic digestion of whole antibodies is necessary. Here, we applied the procedure published by Sharma et al. [63], incorporating the tris(2-carboxyethyl) phosphine hydrochloride (TECP) compound for whole antibodies, cleaving the disulphide bridges. After reducing the antibodies, the half region of F_c_ with the half region of F_ab_ (heavy constant CH and variable VH chains) and the half region of F_ab_ (light constant CL and variable VL chains) were obtained (Figure 1). In this approach, the separation of products from the reaction mixture is not required. The immunosensor fabrication consists of the following steps (Figure 2): (i) TBBT SAM deposition on the Au electrode; (ii) covalent deposition of AuNPs; (iii) covalent deposition of the half-antibody fragment; and (iv) filling of empty free spaces and eliminating of unspecific binding by BSA. Each step of the modifications of the working Au electrode was evaluated by both the performed electrochemical techniques: CV and EIS in the presence of K_3_[Fe (CN)_6_]/K_4_[Fe (CN)_6_] (1:1) as a redox marker, with a 0.5 mM concentration, and using 0.1 M PBS as an electrolyte at a stable pH of 7.4. The CV and EIS curves recorded after each step of the modification are presented in Figure 5. The potential separation (ΔE_p_) between the oxidation and reduction peaks equal to 86 ± 9 mV recorded for the pure Au electrode confirmed the good reversibility of the redox marker process and cleanness of the surface. This conclusion was also confirmed by the EIS recorded for the bare electrode (Figure 5C and Table 1). The straight line is related to the diffusion-controlled electrochemical process. 

However, after the TBBT SAM deposition, the surface of the working electrode was blocked. This caused a decrease in the accessibility of the ferro- and ferricyanides [Fe (CN)_6_]^3−/4−^ redox marker to the interface with the working electrode surface. Consequently, the oxidation and reduction peaks separation increased to 386 ± 44 mV and the electron transfer resistance (R_et_), estimated with EIS, was 560 ± 58 kΩ. Both parameters confirmed the successful deposition of TBBT SAM. After the immobilization of the AuNPs, the difference between the oxidation and reduction peak potential decreased to the value 168 ± 18 mV (Figure 5A). Correspondingly, the charge transfer resistance was reduced to 120 ± 11 kΩ (Figure 5B and Table 1). This result indicated the presence of the quasi-reversible electron transfer mechanism between the transducer and redox marker [86], and subsequently enhanced to form a higher electric conductivity interface similar to the bare gold electrode. 

The immobilization of the half-IgG caused a substantial decrease in electrode reversibility. The CV peaks separation increased to 358 ± 76 mV for AuNPs. CV data were also confirmed by EIS. After immobilization of half-IgG, the charge transfer resistance increased to 256 ± 35 kΩ in the case of AuNPs. These parameters confirmed the successful deposition of half-IgG on the nanoparticles. The filling of empty space and blocking unspecific binding with BSA caused the additional reversibility to decrease. The final value charge transfer resistance of the immunosensor based on AuNPs was 563 ± 76 kΩ. However, the peaks of the oxidation and reduction process were not clearly distinguishable (ΔEp = 404 ± 35 mV). This result confirmed the complete modification of the sensing platform, which, after that, was ready for the detection of the collagen I analyte.

### 3.4. EIS-Based Collagen Type I Immunosensing

The sensing of collagen I using the prepared immunosensors was carried out with Electrochemical Impedance Spectroscopy (EIS). The cyclic voltammetry (CV) technique was not suitable because of its high irreversibility. The measurements of the sensitivity of the performed immunosensors were conducted with concentrations of 1, 2, 3, 4 and 5 pg/mL of collagen type I in PBS, pH 7.4. The interactions between half-IgG and collagen I impeded the accessibility of the ferro- and ferricyanides to the surface of the transducer and, consequently, increased the electron transfer resistance (Ri). These values were obtained by fitting the EIS spectra using the circuit model presented in Figure 6. 

To obtain the appropriate calibration curves, the relative changes in electron transfer resistance (ΔR) were expressed using the following equation [51]:(1)$$\Delta R=\frac{R_{i}-R_{0}}{R_{0}}\times 100\%$$
where R_0_ represents the value of the electron transfer resistance of the sensing system recovered in the 0.1 M PBS buffer without application of the analyte. The values of the relative changes in electron transfer resistance increased proportionally with higher concentrations of collagen I for the studied system (Figure 7). The slope of the calibration curve and the range of standard deviations determined the precision sensing of collagen I. 

The limit of detection (LOD) was determined by using the following formula [87]:
(2)$$\mathrm{LOD}\;=\;\frac{3.3\sigma }{S}$$
where σ is the value of the standard deviation for the y-intercept and S represents a slope of the regression line. The determined value of LOD for the collagen immunosensor was 0.38 pg/mL. A selectivity study was performed using bovine serum albumin (BSA), with a concentration range from 1 to 5 pg/mL as a control. These compounds revealed negligible responses towards the presented sensing platform, and the accurate selectivity properties were also confirmed. According to the obtained results, the performance of the collagen immunosensor was validated in the dynamic range from 1 to 5 pg/mL, which was evaluated previously by the SPR measurements. The calculated LOD is only a theoretical value, indicating the minimum concentration of the collagen type that is possible to detect using the performed system. 

However, the SPR angle shift for the higher concentration above the dynamic range was not accurate for the further determination of the collagen type I protein. Therefore, this sensor has an important application in the detection of a very low range of collagen content, which is required for future samples taken from patients to confirm the presence of the initial stage of collagen synthesis. The verification of the collagen type I concentration in the picomolar range has a significant impact on the initial and rapid diagnosis of the regeneration mechanism of tendons and ligaments [13,15,17,18]. Accordingly, it allows selecting the appropriate treatment, while also revealing the potential capability of supporting the healing of patients by injection with collagen type I/III and hyaluronic acid, or by applying invasive surgery [10,16].

## 4. Discussion

According to the obtained results, the limit of detection demonstrated that the performed immunosensor based on the nano-Au particles had a remarkable high sensitivity range for collagen I detection, which was caused by many factors, such as the high electrical conductivity of the AuNPs with directly immobilized, complementary half-IgG antibodies. Additionally, the application of the AuNPs increased the surface area of the transducer [46,47,48,49]. This directly allowed for combining higher amounts of the half-IgG antibody receptors [51,54], which had a crucial role in the sensitivity of the immunosensor. Therefore, these kinds of modifications based on covalent bonding from each side between Au and S can definitely guarantee a better and more stable SAM layer compared to the other immobilization systems based on the G or A proteins [31,32,33,34,35], which have relatively higher resistivity and, consequently, lower electrical signals towards the covalent immobilization of receptors through metal nanoparticles. Moreover, there are various methods for the determination of the collagen content, as summarised in Table 2. One of the most common techniques is ELISA; however, this enables obtaining a minimum detection limit in the range of approximately 1 ng/mL. Subsequently, the application of electrochemical immunosensors based on both techniques, Electrochemical Spectroscopy Impedance, and Surface Plasmon Resonance, allows for achieving significantly higher sensitivity, in the range of picograms per millilitre, against the previous commonly used techniques, such as ELISA.

Additionally, the fabrication of immunosensors presented in this research is rather simple. The important advantage of this platform ensures high and strong stability of the antibody immobilization using covalent bonds against to the weaker electrostatic interactions. Additionally, an independent examination through Surface Plasmon Resonance confirmed this efficiency in collagen detection in the significantly low pg/mL concentration range by the presented immunosensor. However, the electrochemical sensing analyser [96,97] costs much less in comparison to spectroscopic devices [94]. 

Furthermore, relatively low sample consumption at the µL level enhances the attractiveness of the presented immunosensors. Finally, the portable electrochemical device can be a potential solution to directly use for the medical diagnostic system without specific and expensive ELISA kit assays.

## 5. Conclusions

The results obtained proved the suitability of the 4,4′-thiobisbenzenethiol SAM formed on the Au electrode for the efficient covalent immobilization of AuNPs. The determined value of the limit of detection was 0.38 pg/mL, which is a remarkably low value in comparison to the current collagen sensors. In the presence of the control compound (BSA), very moderate responses were generated, and this proved the selectivity properties of the presented immunosensor.

The proposed construction of the immunosensor with a nano-Au SAM layer has several significant advantages, such as the high stability of the receptor layer, using covalent bonds through the enzymatically reduced half-IgG antibody, and additionally increasing the surface area of the electrode, which directly supported to immobilize more antibodies. Therefore, the presented immunosensors have improved on the currently used methods by being able to validate the presence of a low concentration of collagen type I, without requiring a large volume of the analysed sample. Overall, the presented immunosensors have the potential to make a great impact on future applications in medical laboratory diagnosis, based on the pg/mL level of sensitivity, good selectivity, very small sample consumption and simple fabrication method at a reasonable cost.

## Figures and Tables

**Figure 1 biosensors-11-00227-f001:**
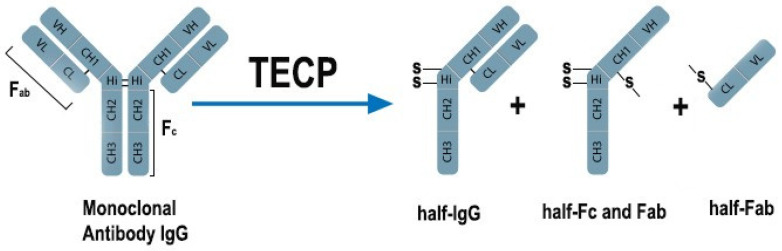
Llustration of the enzymatic reduction of the monoclonal antibody IgG by the disulphide-bridge-bond-breaker TECP (tris (2-carboxyethyl) phosphine hydrochloride) [63].

**Figure 2 biosensors-11-00227-f002:**
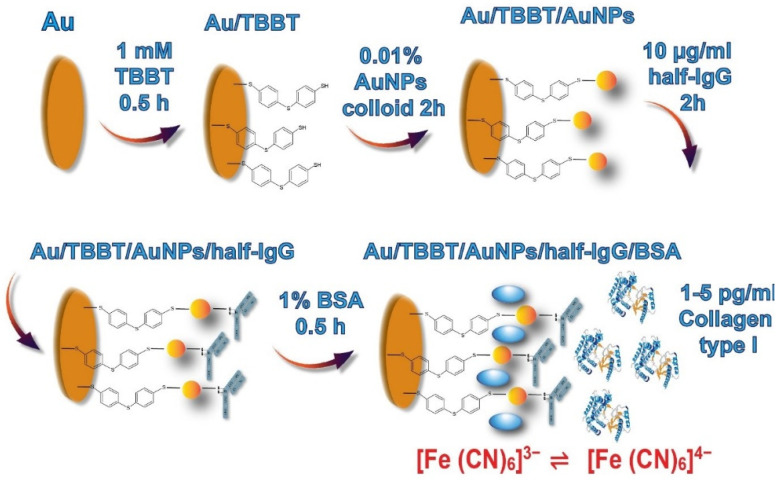
Illustration of the enzymatic reduction of the monoclonal antibody IgG by the disulphide-bridge-bond-breaker TECP (tris(2-carboxyethyl) phosphine hydrochloride) [63].

**Figure 3 biosensors-11-00227-f003:**
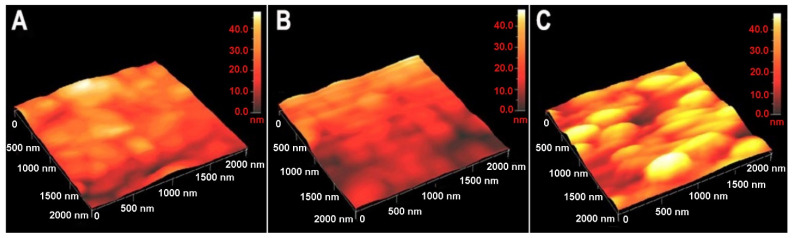
AFM 3D images (2 μm × 2 μm) of (**A**) the mica substrate coated with a thin film of Au (111), (**B**) the TBBT monolayer, and (**C**) the AuNP-modified Au surface.

**Figure 4 biosensors-11-00227-f004:**
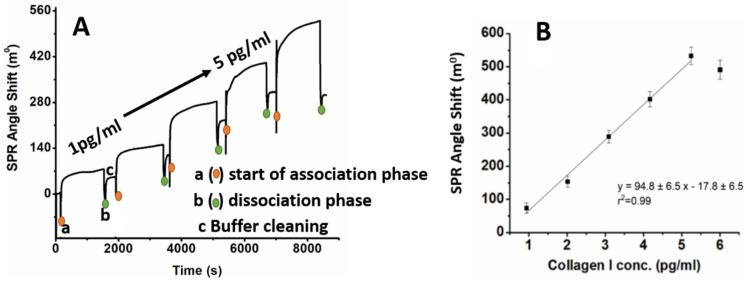
(**A**) SPR response, and (**B**) calibration curve of Au/TBBT/AuNPs/half-IgG/BSA towards collagen I in the concentration range from 1–5 pg/mL in PBS buffer.

**Figure 5 biosensors-11-00227-f005:**
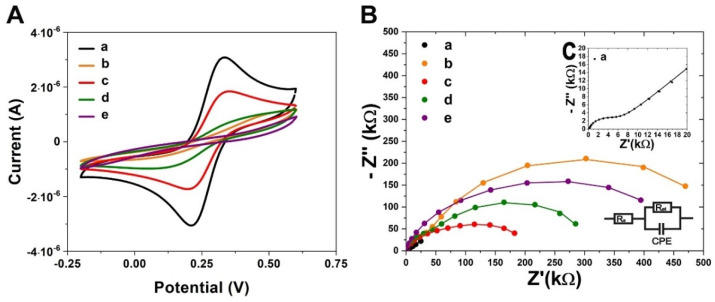
(**A**) Cyclic voltammetry (CV) curves, and (**B**,**C**) electrochemical impedance spectra (EIS) recorded after each step of the Au electrode modifications: (a) bare Au electrode; (b) TBBT SAM deposition; (c) immobilization of the metal nanoparticle colloids AuNPs; (d) immobilization of half-IgG; (e) deposition of BSA. Measuring conditions of EIS: applied frequencies from 0.01 Hz —100 kHz with the bias potential at 0.17 V in 10 mV of ac amplitude. The EIS spectra were fitted to the Nyquist plots for an electrical circuit model, which included R_s_—solution resistance; R_et_—electron transfer resistance; and CPE—constant phase element. Composition of used electrolyte: 0.1 M PBS, pH 7.4 with 0.5 mM K_3_[Fe (CN)_6_]/K_4_[Fe (CN)_6_] (1:1).

**Figure 6 biosensors-11-00227-f006:**
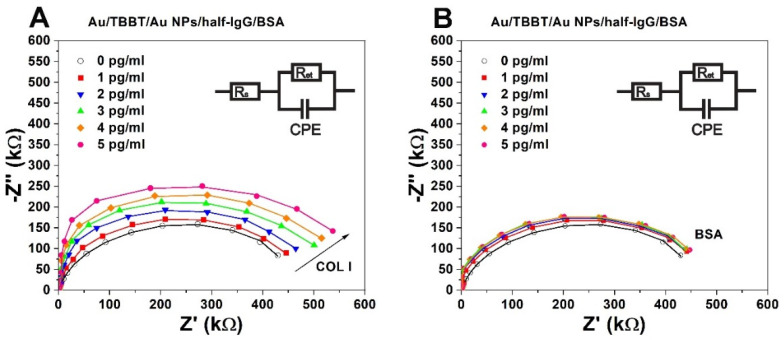
Electrochemical impedance spectra: (**A**) Au/TBBT/AuNPs/half-IgG/BSA in the presence of collagen I, (**B**) Au/TBBT/AuNPs/half-IgG/BSA in the presence of BSA. Concentration range of collagen I analyte: (○) 0, (■) 1, (▼) 2, (▲) 3, (♦) 4, (●) 5 pg/mL. Measuring conditions of the EIS: applied frequencies from 0.01–100 kHz with the bias potential at 0.17 V in 10 mV of ac amplitude. The EIS spectra were fitted to the Nyquist plots for an electrical circuit model, which included R_s_—solution resistance, R_et_—electron transfer resistance, and CPE—constant phase element. Composition of the used electrolyte: 0.1 M PBS, pH 7.4 with 0.5 mM K_3_[Fe (CN)_6_]/K_4_[Fe (CN)_6_] (1:1).

**Figure 7 biosensors-11-00227-f007:**
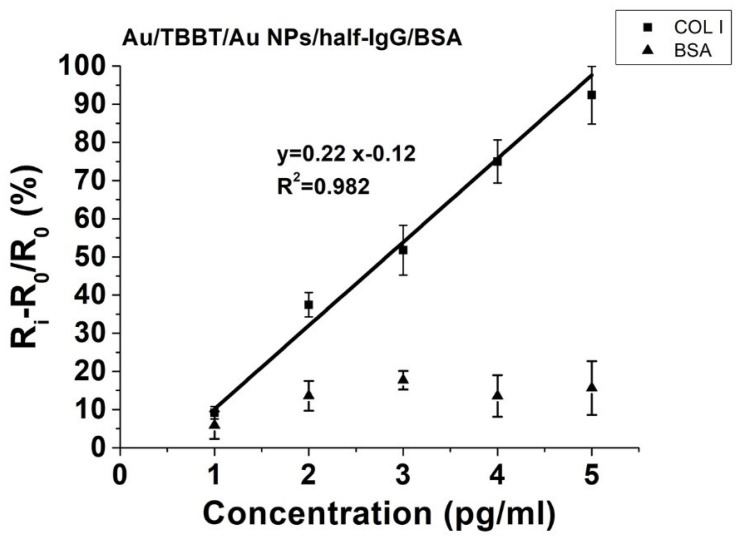
The correlation between (R_i_ − R_0_)/R_0_ (%) and the applied range of collagen I (COL I) concentrations (pg/mL) and the negative control of bovine serum albumin (BSA), as measured for the Au/TBBT/AuNPs/half-IgG/BSA immunosensor system. R_0_ represents the value of the electron transfer resistance of a completely functionalised working electrode without any addition of the collagen I analyte (COL I), and R_i_ is the electron transfer resistance after all steps of the electrode functionalisation, which were measured in 0.1 M PBS solution for the range of concentrations from 1–5 pg/mL (*n* = 5) of collagen I (COL I).

**Table 1 biosensors-11-00227-t001:** Summary of the fitting results from the EIS plots through the Randles electrical circuit.

Samples	R_et_ (Ω)	R_s_ (Ω)	CPE (F)
Au/TBBT	560 × 10^3^	13	1.07 × 10^−4^
Au/TBBT/AuNPs	120 × 10^3^	7.4	2.46 × 10^−4^
Au/TBBT/AuNPs/half-IgG	256 × 10^3^	9.7	1.28 × 10^−4^
Au/TBBT/AuNPs/half-IgG/BSA	563 × 10^3^	16	1.71 × 10^−4^

**Table 2 biosensors-11-00227-t002:** Comparison of the different techniques for the determination of the collagen content.

Method of Measurement	Type of Collagen	Type of Sample	Detection Limit	Range of Detection	References
ELISA	collagen I	synthetic C peptide fragments of type I collagen (α−1)	10 ng/mL	10–1000 ng/mL	[88]
Fluorescence Assays	collagen triple helix GPO	synthetic collagen triple helix GPO	30 nM	100–1000 nM	[89]
Surface Plasmon Resonance Imaging (SPRI)	collagen IV	human blood plasma	2.4 ng/mL	10–1000 ng/mL	[90]
ELISA	collagen I	human serum	0.83 ng/mL	0.83–500 ng/mL	[91]
ELISA	collagen III	human serum	0.6 ng/mL	0.9–200 ng/mL	[92]
Localized Surface Plasmon Resonance (LSPR)	collagen IV	extracted collagen IV from human placenta	10 ng/mL	10–1000 ng/mL	[19]
ELISA	collagen I	human blood plasma	5.3 pg/mL	40–2500 pg/mL	[93]
Electric Field-Induced Accumulation	collagen I	extracted collagen I from human placenta Bornstein and Traub type I	3.0 pg/mL	3–60 pg/mL	[94]
Cyclic Voltammetry (CV)	collagen I	solution of synthetic collagen I	0.5 pg/mL	0.5 pg/mL–0.5 ng/mL	[95]
EIS (AuNPs transducer)	collagen I	bovine collagen solution I	0.38 pg/mL	1.0–5.0 pg/mL	Present work

## Data Availability

All presented data from this research are available on request to the corresponding authors.

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
