# Peer review of "Impedimetric and Plasmonic Sensing of Collagen I Using a Half-Antibody-Supported, Au-Modified, Self-Assembled Monolayer System"

_biosensors, 2021, doi:10.3390/bios11070227_

Round 1
Reviewer 1 Report
In this paper,lots of sentences are based on an oral statement (such as 3 sentences in first paragraph, 1. Therefore, it is necessary to construct and apply wearable sensors or in situ biomarkers,… 2. it is necessary to indicate the concentration of collagen… 3. Therefore, it is vital to implement sensitive systems suitable for… 4. Therefore, it is essential to modify the surface of the working…), we suggest author need a “Extensive editing of English language and style required”.
Author Response
Reviewer #1
In this paper lots of sentences are based on an oral statement (such as 3 sentences in first paragraph,
Remark: Authors do appreciate every comment and suggestion of reviewer. This revised version of manuscript addresses quesires raised by reviewers.
Comment 1. Therefore, it is necessary to construct and apply wearable sensors or in situ biomarkers,…
Response: As suggested, the corrections have been made in the revised manuscript.
‘Accordingly, it is highly demanded to construct and apply wearable sensors or in situ biomarkers…’
Comment 2. it is necessary to indicate the concentration of collagen…
Response: As suggested, the corrections have been made in the revised manuscript.
‘To increase the retreatment process, it is significantly relevant to indicate the concentration of collagen type I with which directly enhances the regeneration of damaged tendon tissue.’
Comment 3. Therefore, it is vital to implement sensitive systems suitable for…
Response: As suggested, the corrections have been made in the revised manuscript.
‘For this reason, it is required to implement sensitive systems suitable for detecting collagen type I with an accurate level of selectivity.’
Comment 4. Therefore, it is essential to modify the surface of the working…),
Response: As suggested, the corrections have been made in the revised manuscript.
‘For this purpose, it is fundamental to modify the surface of the working electrode to enhance the connection with the antibody receptor’
Comment 5. we suggest author need a “Extensive editing of English language and style required”.
Response: As suggested, the manuscript is revised to improve the English style.
Reviewer 2 Report
Comments for authors
biosensors-1245836
Impedimetric and Plasmonic sensing of Collagen I using half antibody supported Au modified self-assembled monolayer system
Overview
The manuscript describes the development of an impedimetric approach to quantify collagen type I, a biomarker related to tendon inflammation which plays an important role in tissue regeneration. The authors used antibody fragments as receptors, which were immobilized onto gold nanoparticle modified electrodes. The electrode modification steps were assessed by electrochemical techniques such as cyclic voltammetry (CV) and electrochemical impedance spectroscopy (EIS), using the redox probe [Fe(CN)6]3-/4- in the supporting electrolyte, as well as AFM and SPR. The signal transduction is the charge transfer resistance (Rct) of a redox probe in buffer (PBS), which allowed the authors to quantify the target in pg/mL levels. Although I believe this manuscript is interesting and fits with the Biosensors’ journal readers, some points need to be addressed in this current version, as detailed below.
Comments and suggestions
- Line 68. It could be interesting the authors discuss that proteins A and G are specially used for orientating antibodies on electrode surfaces (tail-on fashion) since these proteins present a high affinity for the Fc antibody region.[1] Also, it is not clear what the authors mean by “result in high resistance on the electrode layer”. Probably, the authors refer to the insulating feature of these proteins regarding electrochemical electron transfer between a redox species in solution and the electrode, which can compromise the sensitivity of specific electrochemical approaches, such as cyclic voltammetry (CV) and faradaic electrochemical impedance spectroscopy (EIS). I suggest the authors revise this paragraph to provide clear information.
- Line 73. There are different electrochemical approaches that do not require redox species in the sample for target detection. Please, see references[2, 3]
- Line 86. Please, note that the unit Ω.cm is for resistivity.
- Linea 89-91. It is not clear why the electrochemical blocking effect of recognized targets onto surfaces over a redox-active marker can cause changes in optical signals. Please, clarify the sentence.
- Line 98. Despite the authors have cited a specific reference, It could be interesting a brief discussion about the reasons for the non-necessity of the purification process using TCP enzyme for antibody cleavage.
- Line 112. Since the proposed biosensing approach is based on EIS and surface plasmon resonance (SPR), it will be for the benefit of the readers the authors include a brief introduction of the methods, especially highlighting the reasons for the choice of these techniques in this work (is this because they present high sensitivity? Easy to miniaturize? etc.). It is quite important since no discussion was provided in this version.
- Also, specify the sequence of the sizes of the used alumina for the polishing step. Did you rinse the electrode between steps? Please, provide more details.
- Line 137. What do you mean by “activate the Au surface” of the electrode with KOH? Usually, this process is to remove electrochemically any impurities onto the electrode surface by desorption[4]. Please, clarify.
- Line 197. The authors claiming that the roughness decrease after TBBT immobilization. However, the obtained values are too close and, importantly, the error of the measurements precludes such affirmation. Probably, a statistical test (such a t-test) can confirm whether the SAM immobilization causes changes in the electrode morphology. Also, it seems that the discussion by Park et al. relies on thickness measurements rather than roughness, hence the authors could perform a similar analysis from their results.
- Lines 209-212. Since the used nanoparticles have a diameter of approximately 20 nm, what does it mean that the particles in the surface present dimensions of 411 x 168 nm²? Cluster formation?
- Lines 228-230. Please, note that from the submitted material, figure 2S is AFM characterization. Please, update the materials. Also, rewrite the sentence. It has repetitive information and it is confusing.
- Line 233. Please, provide Figure 3S. Also, it is not clear if the authors assessed BSA immobilization by SPR since the authors jumped this step in the discussion.
- Lines 242-243. The discussion regarding the decreasing of SPR signal is quite confusing. The interaction of the target protein with specific sites of receptors is immunoreaction, and it should increase the shift angle in SPR, not decrease (as seen in the different concentrations). The decrease could be due the desorption of non-specific bonded protein onto the surface, or a saturation point that was reached (which obey isotherm models such as Langmuir).
- Line 257-258. The thiol group is attached to the gold nanoparticle surface rather than Au surface. Please, correct in order to avoid misunderstanding.
- Figure 5. It is hard to see some CV curves in part (a). Maybe plotting with different colors could help the readers differentiate the results. The same for Nyquist plots.
- Figures 5b and 6. Please, note that the impedance data of the axes in the Nyquist impedance plot have to present scales in the same size or proportional (i.e. orthonormal)[5] in order to avoid plot distortions. In addition, since the data were fitted using Randles circuit, provide a table in SI with all fitted parameters and include the fitting in all Nyquist plots. Also, include a zoom (or as part c in Figure 5) to show the Nyquist plot for the bare gold electrode.
- Lines 303-304. Please, include a discussion regarding the increase in current/decrease in Rct after gold nanoparticles immobilization. Refer to work [6] and references for details.
- Line 365. Please, note that LOD is calculated using signals from blank not from the lowest target concentration. Another approach is using the standard deviation of the y-intercept.[7] Please, correct accordingly.
- Table 1. Please, include the type of sample used for each approach, since it impacts LOD and dynamic range.
- The dynamic range of collagen quantification is too narrow, from 1 to 5 pg/mL. Ideally, the approach should present at least one order of magnitude for target quantification (as can be seen in the examples presented in Table 1). Consequently, in order to show the feasibility of the method, it could be important to perform more assays in order to explore the dynamic range of collagen quantification by both lower and higher concentrations of collagen (note that SPR data shown a saturation characteristic for concentrations above 5 pg/mL. However, such analysis should be done for EIS as well since they are different techniques and present different sensitivities and transduction mechanisms. Hence, EIS could present a different behavior).
General.
- Line 118. A comma between “methanol” and “potassium hydroxide” is missing.
- Line 137. Typo in the sulfuric acid formula.
- Line 136 and 138. Please, try to keep consistency in the nomenclature over the text. For example, in some parts, it is used “potassium hydroxide”, while others “KOH”.
- Line 179. Typo in “Ausurface”.
- Line 204. Note that is gold nanoparticles, not nanogold particles.
References
- Trilling, A.K., J. Beekwilder, and H. Zuilhof, Antibody orientation on biosensor surfaces: a minireview. Analyst, 2013. 138(6): p. 1619-1627.
- Santos, A., P.R. Bueno, and J.J. Davis, A dual marker label free electrochemical assay for Flavivirus dengue diagnosis. Biosensors and Bioelectronics, 2018. 100: p. 519-525.
- Ben Aissa, S., et al., Design of a redox-active surface for ultrasensitive redox capacitive aptasensing of aflatoxin M1 in milk. Talanta, 2019. 195: p. 525-532.
- Tkac, J. and J.J. Davis, An optimised electrode pre-treatment for SAM formation on polycrystalline gold. Journal of Electroanalytical Chemistry, 2008. 621(1): p. 117-120.
- Orazem, M.E. and B. Tribollet, Electrochemical impedance spectroscopy. 2008, Hoboken, New Jersey: John Wiley & Sons.
- Lin, D., et al., A regenerating ultrasensitive electrochemical impedance immunosensor for the detection of adenovirus. Biosensors and Bioelectronics, 2015. 68: p. 129-134.
- Shrivastava, A. and V. Gupta, Methods for the determination of limit of detection and limit of quantitation of the analytical methods. Chronicles of Young Scientists, 2011. 2(1): p. 21-25.
Author Response
Reviewer 2:
The manuscript describes the development of an impedimetric approach to quantify collagen type I, a biomarker related to tendon inflammation which plays an important role in tissue regeneration. The authors used antibody fragments as receptors, which were immobilized onto gold nanoparticle modified electrodes. The electrode modification steps were assessed by electrochemical techniques such as cyclic voltammetry (CV) and electrochemical impedance spectroscopy (EIS), using the redox probe [Fe(CN)6]3-/4- in the supporting electrolyte, as well as AFM and SPR. The signal transduction is the charge transfer resistance (Rct) of a redox probe in buffer (PBS), which allowed the authors to quantify the target in pg/mL levels. Although I believe this manuscript is interesting and fits with the Biosensors’ journal readers, some points need to be addressed in this current version, as detailed below.
Comment 1. Line 68. It could be interesting the authors discuss that proteins A and G are specially used for orientating antibodies on electrode surfaces (tail-on fashion) since these proteins present a high affinity for the Fc antibody region.[1] Also, it is not clear what the authors mean by “result in high resistance on the electrode layer”. Probably, the authors refer to the insulating feature of these proteins regarding electrochemical electron transfer between a redox species in solution and the electrode, which can compromise the sensitivity of specific electrochemical approaches, such as cyclic voltammetry (CV) and faradaic electrochemical impedance spectroscopy (EIS). I suggest the authors revise this paragraph to provide clear information.
Response: According to the above comment, the Authors improved this section with a more specific explanation regarding the availability to control the appropriate on-tail orientation of the antibody through the intermediate A and G proteins with corresponded references. Moreover, we specified the information of the electric insulating properties of those proteins (Zhang, B.; Song, W.; Pang, P.; Lai, H.; Chen, Q.; Zhang, P.; Lindsay, S. Role of contacts in long-range protein conductance. Proc. Natl. Acad. Sci. U. S. A. 2019, doi:10.1073/pnas.1819674116), which cause the higher electron transfer resistance through the electrochemical impedance spectroscopy (EIS) experiment.
‘The application of G and A proteins is highly efficient for non-covalently binding antibodies and also providing to orientate them on-tail[28]. Hence, those intermediate proteins have five and two specific domains, which allow to appropriately terminate with the crystallizable region (Fc) and also support to orientate on-tail antibody [29]. That ensures to obtain uniform arrangement of the receptor of their antigen-binding fragment (Fab) against the complementary antigen and consequently increase the sensitivity properties.[30] Nevertheless, the high molecular weights of the intermediate proteins and their insulating electric properties [31]between the redox marker and transducer substrate have a negatively influences electrochemical sensor performance and they reveal higher electron transfer resistance through the Electrochemical Impedance Spectroscopy (EIS).’
Comment 2. Line 73. There are different electrochemical approaches that do not require redox species in the sample for target detection. Please, see references[2, 3]
Response: Authors have taken into account this suggestion and incorporated in the introduction section those examples of the electrochemical sensing platforms without any additional redox marker with corresponding references.
‘Furthermore, there are sort of different attempts allow for biosensing excluded additional redox species [36] such as label-free impedance derived redox capacitance for Flavivirus dengue detection [37].’
Comment 3. Line 86. Please, note that the unit Ω.cm is for resistivity.
Response: Authors corrected this unit, and changed it for the SI unit of Ω.cm.
‘Application of the SAM gold nanoparticles reveals high electrical conductivity and low resistance (2.44 Ω ∙cm) [54,55], which consequently improved the analytical parameters of the immunosensors and sensitivity properties.’
Comment 4. Linea 89-91. It is not clear why the electrochemical blocking effect of recognized targets onto surfaces over a redox-active marker can cause changes in optical signals. Please, clarify the sentence.
Response: Authors rewrote this section and clarify that the electrochemical blocking effect of the decreasing accessibility to the redox marker corresponds only to the electrochemical impedance spectroscopy studies (EIS) without any additional amplification mechanism, but the effect of the highly orientated antibodies allow to generate sufficient plasmonic resonance effect and shift SPR angle using the other optical method.
‘Application of the system-based orientated half-reduced antibodies allows obtaining a highly specific immunosensor to the target protein. The recognition of the analyte contributes to the decreasing of the accessibility of the re-dox-active marker registered through electrochemical impedance spectroscopy (EIS). Analogically, the highly sensitive orientation of half-antibody contributes to the efficient of the interaction with complementary antigen, induce the mechanism of the plasmonic resonance and allow to detection of the shift SPR angle. Consequently, it is not required to use any additional amplification mechanism to verifying changes of the optical or electrochemical signals towards various concentrations of the analyte.’
Comment 5. Line 98. Despite the authors have cited a specific reference, It could be interesting a brief discussion about the reasons for the non-necessity of the purification process using TCP enzyme for antibody cleavage.
Response: Authors rephrased and improved that the additional process of the half-antibody purification was unnecessary because of their function in blocking empty spaces onto the receptor layer.
‘In this approach, the further separation process of the reaction mixture was not necessary because the other fractions of the half-Fc and Fab with half-Fab antibody fragments contribute in the mechanism of the efficient blocking empty spaces onto the receptor layer.’
Comment 6. Line 112. Since the proposed biosensing approach is based on EIS and surface plasmon resonance (SPR), it will be for the benefit of the readers the authors include a brief introduction of the methods, especially highlighting the reasons for the choice of these techniques in this work (is this because they present high sensitivity? Easy to miniaturize? etc.). It is quite important since no discussion was provided in this version.
Response: As suggested, this section has been improved and briefly introduced the main features of the conducted two independent techniques, electrochemical (EIS) and optical (SPR) with the potential application as a universal and miniaturized system, which can also support further mass production.
‘This universal system can allow to quickly verify the content of the collagen I with the miniaturized portable electrochemical device based on the evaluation of the accessibility of the redox marker to the sensor surface through the EIS measurement of the electron transfer resistance corresponding to the appropriate value of the analyte concentration. Moreover, it ensures to apply the same biosensing system using the independent optical technique, where the interaction between the specific half-reduced antibody and antigen generates through the excitation of the plasmonic effect the shift of the recorded SPR angle. This duality of the verification of the collagen I content support to apply it as a multipurpose system and give the ability to translate for other biosensing platforms.’
Comment 7. Also, specify the sequence of the sizes of the used alumina for the polishing step. Did you rinse the electrode between steps? Please, provide more details.
Response: Authors have added more details regarding the polishing using alumina slurries regarding the order of applied slurries 0.3 μm and 0.05 μm and washing between and after it using methanol and Mili-Q water.
‘The electrodes were polished using alumina slurries (0.30 μm) for 15 min followed by gentle washing with methanol and Milli-Q water. Then, this procedure was conducted again, but using 0.05 μm of alumina slurries to obtain smooth gold surface following rinsing electrodes by methanol and Milli-Q water.’
Comment 8. Line 137. What do you mean by “activate the Au surface” of the electrode with KOH? Usually, this process is to remove electrochemically any impurities onto the electrode surface by desorption [4]. Please, clarify.
Response: Authors have clarified this stage of the electrode preparation in the main manuscript. The process of KOH gold electrode activation is corresponded to the pretreatment of the gold surface and removing any residual impurities absorbed on the transducer surface before the modification.
‘Eventually, the Au surface of the working electrode was pre-treated by the activation through the application of 10 CV cycles in 0.5 M KOH solution. This stage ensured to remove any residual impurities absorbed on the gold electrode surface.’
Comment 9. Line 197. The authors claiming that the roughness decrease after TBBT immobilization. However, the obtained values are too close and, importantly, the error of the measurements precludes such affirmation. Probably, a statistical test (such a t-test) can confirm whether the SAM immobilization causes changes in the electrode morphology. Also, it seems that the discussion by Park et al. relies on thickness measurements rather than roughness, hence the authors could perform a similar analysis from their results.
Response: As suggested, authors checked the roughness values using ANOVA posthoc Tukey statistical test and it was not revealed significant changes regarding the roughness; however, it was revealed the further analyzation the significant changes for the thickness between Au/TBBT and Au/TBBT with AuNPs SAM layer, which is corresponding to the results previously published by Park et al.
‘Obtained roughness 19.4 ± 3.6 nm and thickness 21.05 ± 6.3 nm after the immobilization of TBBT on the Au surface (Fig. 3B) was not significantly changed and it was in the same range in relation to the plain gold surface. was found to decrease up to 19.4 ± 3.6 nm after the immobilization of TBBT on the Au surface (Fig. 3B). After the incorporation of Au nanoparticles, the topography of the TBBT modified surface again changed and attached particles became visible in the corresponding AFM images. The roughness of the modified surfaces changed to 22.6 ± 3.8 nm and the measured thickness significantly increased to 27.6 ± 4.3 nm in comparison to the Au/TBBT sample for AuNPs (Fig. 3C), which is corresponded to similar research performed by Park W. et al. [60] for SAM AuNPs layer Au self-assembled monolayer.
Comment 10. Lines 209-212. Since the used nanoparticles have a diameter of approximately 20 nm, what does it mean that the particles in the surface present dimensions of 411 x 168 nm²? Cluster formation?
Response: The gold nanoparticles have a tendency to aggregate together and form a larger cluster formation with much larger dimensions, those studies have incorporated in the figure 2S at supplementary material. We have improved the results section to explained the presence of the cluster formation.
‘ The average length and breadth of the particles for AuNPs modified Au surface were 411.5 ± 35.6 nm and 168.6 ± 38.7 nm, respectively. These results indicate the formation of the larger aggregated cluster of the combined together gold nanoparticles, which are arrangement with high packed density onto the surface of the electrode.’
Comment 11. Lines 228-230. Please, note that from the submitted material, figure 2S is AFM characterization. Please, update the materials. Also, rewrite the sentence. It has repetitive information and it is confusing.
Response: Authors have corrected this information to an appropriate number of supplementary figure 3S
‘After the deposition of AuNPs on the Au surface the angle shifted from 100 to 1000 and later after immobilization of half-IgG antibodies (Au/TBBT/AuNPs/half-IgG) the angle shifted to 1550 as show in Figure 3S.’
Comment 12: Line 233. Please, provide Figure 3S. Also, it is not clear if the authors assessed BSA immobilization by SPR since the authors jumped this step in the discussion.
Response: Authors apologies for making this mistake. Figure 3S has been added in the supplementary section, we changed the abbreviation of the analysis of the evaluation of changing SPR angle shift for the entirely prepared biosensing platform after BSA deposition Au/TBBT/AuNPs/half-IgG/BSA.
‘The Au/TBBT/AuNPs/half-IgG/BSA immunosensor shows the linear range between 1-5 pg/ml beyond this it shows a decrease in the SPR angle shift.’
Comment 13: Lines 242-243. The discussion regarding the decreasing of SPR signal is quite confusing. The interaction of the target protein with specific sites of receptors is immunoreaction, and it should increase the shift angle in SPR, not decrease (as seen in the different concentrations). The decrease could be due the desorption of non-specific bonded protein onto the surface, or a saturation point that was reached (which obey isotherm models such as Langmuir).
Response: The SPR signal increased consistently upon the exposure of collagen I up to 5 pg/ml, after that response signal decrease. This decrease in the signal is observed due to the saturation after 5 pg/ml. It should be noted that the instrument was able to detect the antibody complex when the increase of its concentration was several orders of magnitude lower, since anti-collagen is a molecule with quite a high molecular weight (130 kDa).
Comment 14. Line 257-258. The thiol group is attached to the gold nanoparticle surface rather than Au surface. Please, correct in order to avoid misunderstanding.
Response: In this paragraph, we mentioned the bond between the thiol group and the main gold polycrystalline substrate. As we explained during the introduction section and visually in Figure 2., the application of TBBT has two sides of the thiol groups, where one is attached to the Au gold surface from one side and the other one is directly linked with gold nanoparticles (AuNPs) using covalent bond for both cases.
Comment 15: Figure 5. It is hard to see some CV curves in part (a). Maybe plotting with different colours could help the readers differentiate the results. The same for Nyquist plots.
Response: As suggested Figure 5. has been changed with different colors of the CV curves and the Nyquist plot for EIS spectra with fitting.
Comment 16. Figures 5b and 6. Please, note that the impedance data of the axes in the Nyquist impedance plot have to present scales in the same size or proportional (i.e. orthonormal) [5] in order to avoid plot distortions. In addition, since the data were fitted using Randles circuit, provide a table in SI with all fitted parameters and include the fitting in all Nyquist plots. Also, include a zoom (or as part c in Figure 5) to show the Nyquist plot for the bare gold electrode.
Response: Authors changed the Figure 5 and 6 accordingly. Authors changed the scale for both x and y-axis as orthogonal for all of the presented EIS graphs with the same range of scale to eliminate the graph distortions. Moreover, included the table with fitting results to the Randles electrical circuit for the calculated value for Ret, Rs and CPE with the standard SI units after each step of the electrode modification.
Comment 17. Lines 303-304. Please, include a discussion regarding the increase in current/decrease in Rct after gold nanoparticles immobilization. Refer to work [6] and references for details.
Response: Auhors have improved this explanation regarding the higher electric conductivity through the restoring quasi-reversible electron transfer mechanism between the redox marker and electrode, which similarly occurred on the bare gold electrode and it consequently promotes the electron transfer and lower electron transfer resistance
‘Correspondingly, the charge transfer resistance was reduced to 120.4 ± 10.5 kΩ (Figure 5B). This result indicated the presence of the reversible electron transfer mechanism between the transducer and redox marker [74], and subsequently enhanced to form higher electric conductivity interface similarly to the bare gold electrode.’
Comment 18. Line 365. Please, note that LOD is calculated using signals from blank not from the lowest target concentration. Another approach is using the standard deviation of the y-intercept.[7] Please, correct accordingly.
Response: Authors recalculated LOD value 0.38 pg/ml using this time the obtained standard deviation of the y-intercept (0.025) and slope (0.22) from the linear regression and consequently correct in the manuscript.
Comment 19. Table 1. Please, include the type of sample used for each approach, since it impacts LOD and dynamic range.
Response: As suggested authors included in the table the new column with the corresponded types of the used sample for each experiment with appropriate references (Table 2.)
Comment 20. The dynamic range of collagen quantification is too narrow, from 1 to 5 pg/mL. Ideally, the approach should present at least one order of magnitude for target quantification (as can be seen in the examples presented in Table 1). Consequently, in order to show the feasibility of the method, it could be important to perform more assays to explore the dynamic range of collagen quantification by both lower and higher concentrations of collagen (note that SPR data shown a saturation characteristic for concentrations above 5 pg/mL. However, such analysis should be done for EIS as well since they are different techniques and present different sensitivities and transduction mechanisms. Hence, EIS could present a different behaviour).
Response: Initially, we used the optical method of the SPR in order to verify the range of the collagen concentration until the threshold of the maximum saturation of the electrode for 5 pg/ml. Accordingly, we have performed the EIS electrochemical measurement as well for the wider range for more than 10 pg/ml, however, the obtained results for the EIS spectroscopy revealed a similar trend, which was observed previously using an optical technique, where the working electrode was saturated after application higher concentration of collagen above 5 pg/ml and it was not recorded the linear increase of the values of electron transfer resistance in relation to the blank sample. Consequently, we determined the range of the selectivity for our immunosensor in this narrow range and these results are coherent to data of the angle shift in the SPR response. Collagen I as a long-chain protein with high molecular weight was immediately detected by the complementary highly specific half-reduced antibody and it plays an important role with quick and efficient saturation of our immunosensor in the narrow range of the applied concentration. Accordingly, we incorporated the EIS data corresponded to the SPR results in this range of the collagen concentration, where it was possible to obtain the linear relation of the sensitivity of performed biosensing platform.
General
Comment 1. Line 118. A comma between “methanol” and “potassium hydroxide” is missing.
Response: As suggested, the corrections have been made in revised manuscript
‘Methanol, potassium hydroxide, ethanol and sulfuric acid were supplied from POCh (Poland).’
Comment 2. Line 137. Typo in the sulfuric acid formula.
Response: As suggested, the corrections have been made in revised manuscript
‘Thereafter, the Au electrode was again electrochemically cleaned using a 0.5 M H2SO4 solution.’
Comment 3. Line 136 and 138. Please, try to keep consistency in the nomenclature over the text. For example, in some parts, it is used “potassium hydroxide”, while others “KOH”.
Response: As suggested authors have changed this issue in the manuscript to keep the consistency in used terminology.
‘CV cycles (no.=100) were performed by immersing the working electrode in 0.5 M KOH solution with applied potential from 0.4 V to -1.2 V using 0.1 V/s of the scan rate.’
Comment 4. Line 179. Typo in “Ausurface”.
Response: As suggested, the corrections have been made in the revised manuscript.
‘Bare Au surface, TBBT modified Au surface, AuNPs attached TBBT modified Au surfaces were analyzed immediately after preparation.’
Comment 5. Line 204. Note that is gold nanoparticles, not nanogold particles.
Response: As suggested, the corrections have been made in the revised manuscript.
‘Moreover, the diameter of used gold nanoparticles is commonly determined for the colloid solution by the dynamic light scattering (DLS) technique.’
References
- Trilling, A.K., J. Beekwilder, and H. Zuilhof, Antibody orientation on biosensor surfaces: a minireview. Analyst, 2013. 138(6): p. 1619-1627.
- Santos, A., P.R. Bueno, and J.J. Davis, A dual marker label free electrochemical assay for Flavivirus dengue diagnosis. Biosensors and Bioelectronics, 2018. 100: p. 519-525.
- Ben Aissa, S., et al., Design of a redox-active surface for ultrasensitive redox capacitive aptasensing of aflatoxin M1 in milk. Talanta, 2019. 195: p. 525-532.
- Tkac, J. and J.J. Davis, An optimised electrode pre-treatment for SAM formation on polycrystalline gold. Journal of Electroanalytical Chemistry, 2008. 621(1): p. 117-120.
- Orazem, M.E. and B. Tribollet, Electrochemical impedance spectroscopy. 2008, Hoboken, New Jersey: John Wiley & Sons.
- Lin, D., et al., A regenerating ultrasensitive electrochemical impedance immunosensor for the detection of adenovirus. Biosensors and Bioelectronics, 2015. 68: p. 129-134.
- Shrivastava, A. and V. Gupta, Methods for the determination of limit of detection and limit of quantitation of the analytical methods. Chronicles of Young Scientists, 2011. 2(1): p. 21-25.
Reviewer 3 Report
This manuscript constructed an electrochemical immunosensor for collagen I detection using a self-assembled monolayer (SAM) of the gold nanoparticles (AuNPs) and covalently immobilized half-reduced monoclonal antibody as a receptor, the limit of detection (LOD) of 0.43 pg/ml, significantly lower than the traditional methods. Overall, the method for the immunosensor development is novel, innovative, and the experimental procedures and results are well organized and written. However, there are several questions should be addressed.
- Though the author declare that the separation of half-IgG and Fab parts from the reaction mixture was not necessary, how do you ensure that the entire antibody molecule is converted to half-IgG and Fab?
- If the immunosensor can be evaluated with clinical samples, the sensitivity results will be more reliable and more meaningful.
- Line71: Need to cite references.
- Line137: H2SO¬4need to be corrected
Author Response
Reviewer #3
This manuscript constructed an electrochemical immunosensor for collagen I detection using a self-assembled monolayer (SAM) of the gold nanoparticles (AuNPs) and covalently immobilized half-reduced monoclonal antibody as a receptor, the limit of detection (LOD) of 0.43 pg/ml, significantly lower than the traditional methods. Overall, the method for the immunosensor development is novel, innovative, and the experimental procedures and results are well organized and written. However, there are several questions should be addressed.
Comment 1. Though the author declare that the separation of half-IgG and Fab parts from the reaction mixture was not necessary, how do you ensure that the entire antibody molecule is converted to half-IgG and Fab?
Response: Authors followed the protocol described by the Sharma et al. (Sharma, H.; Mutharasan, R. Half antibody fragments improve biosensor sensitivity without loss of selectivity. Anal. Chem. 2013, doi:10.1021/ac3035426.). It was adapted this technique to reduced the IgG antibody through the disulfide bridges by enzymatic reaction using TECP compound. We extended our explanation regarding the fact that it was not necessary to purify the received fractions after digestion of the antibody because all of the other fragments such as half-Fc and Fab with half-Fab have a function in blocking empty spaces onto the receptor layer.
‘In this approach, the further separation process of the reaction mixture was not necessary, because the other fractions of the half-Fc and Fab with half-Fab antibody fragments contribute in the mechanism of the efficient blocking empty spaces onto the receptor layer.’
Comment 2. If the immunosensor can be evaluated with clinical samples, the sensitivity results will be more reliable and more meaningful.
Response: In our research, we proposed a quick, highly specific and selective system for precise collagen type I detection. Initially, we have conducted a study using the bovine collagen type I solution in the phosphate buffer saline (PBS buffer). However, in case of the further step of our research, we are planning later to transfer our biosensing system from the blood serum to indicate the collagen content using real-sample. However, the main aim of the presented results was related to demonstrating the strategy of the stable covalent immobilization highly specific half reduced antibodies to detect with high sensitivity and selectivity properties using two independent techniques electrochemical (EIS) and optical (SPR). Accordingly, for this reason, the application of the bovine collagen type I solution was a sufficient and effective method to validate the performance of the constructed immunosensor.
Comment 3. Line71: Need to cite references.
Response: Authors have improved and modified this section with appropriate references regarding higher electron transfer resistance for the high molecular weight of intermediate A and G proteins, which have insulating properties.
‘Common methods are based on ‘lock and key’ approaches, such as G and A proteins, which enable immobilization of the receptor antibody on the surface of the transducer [26,33–36]. The application of G and A proteins is highly efficient for non-covalently binding antibodies and also providing to orientate them on-tail[37]. Hence, those intermediate proteins have five and two specific domains, which allow to appropriately terminate with the crystallizable region (Fc) and also support to orientate on-tail antibody [38]. That ensures to obtain uniform arrangement of the receptor of their antigen-binding fragment (Fab) against the complementary antigen and consequently in-crease the sensitivity properties.[39] Nevertheless, their high molecular weights of the intermediate proteins and their insulating electric properties [40]between the redox marker and transducer substrate have a result in high resistance on the electrode layer, which negatively influences electrochemical sensor performance and they reveal higher electron transfer resistance through the Electrochemical Impedance Spectroscopy (EIS).’
Comment 4. Line137: H2SO¬4 need to be corrected
Response: As suggested, the corrections have been made in the revised manuscript.
‘Thereafter, the Au electrode was again electrochemically cleaned using a 0.5 M H2SO4 solution.
Reviewer 4 Report
The manuscript presents an interesting work on the detection of collagen I using the SPR and EIS biosensors platform. Although the manuscript is of interest, some major corrections - listed below- are recommended prior to process the manuscript further. Please do not feel intimidated by the long list, some are actually minor corrections.
- The Author should explain better why physicians or patients need to detect/monitoring of collagen. This was presented only partially. It would be good to know how knowing a specific concentration of collagen I can help in the treatment plan and how this biosensor can be beneficial over existing methods.
- The Authors should explain why the concentration range they reached is advantageous over the concentration ranges achieved by existing test, such as ELISA. In other words, why the concentration in the pg/mL range is better than the concentration range in ng/mL. Is there any clinical/diagnostic benefit?
- The Authors should update some part of the bibliography, e.g. Gelse et al., 2003, MAffuli et al. 2004, Etherington et al., 1981, etc. These references should be replaced with more recent works.
- The authors claimed that no separation process was needed after the enzymatic digestion of IgG. Please explain how did you reach this conclusion. I was expecting an enzymatic inactivation step/checks.
- Although the authors stated that the half IgG was covalently immobilised on the Au/TBBT/AuNPs, it seems that the immobilisation was achieved via adsorption (lines 142-145). Please clarify.
- Line 157-158: Did the authors use any means to protect the electrodes from photo-degradation?
- Ferro-ferricyanides is usually referred to as the redox-active probe (line166).
- On AFM study result section: Is the difference in the roughness statistically significant? If so, add the statistical analysis, please. the quality of the AFM images need to be improved. Furthermore, please add a possible explanation on roughness decreased value observed after TBBT immobilisation.
- Line 224: Authors claims "SPR technique is the most convenient tool to study the interfacial interaction between the analyte (antigen) and the immobilized biomolecules (antibody) in real-time" ->, please explain/comparison with other tools and evidence of this claim. Else please just rephrased with a more objective statement, such as claiming that "the SPR technique is an analytical tool to study the interfacial interaction...in real-time".
- Figure placeholder mismatch and figure missing. I could not locate Figure 2S (line(230). Please double check.
- I can only see Figure 1S and 2S only in the Supplementary info, and these refer to the AFM study.
- The figure placeholder should be consistent between the main text and the figure caption. E.g., Figure 4 (a) in line 242 and figure caption reported as Figure 4 A.
- It is unclear if the Authors developed a regeneration protocol for the SPR sensor as they stated that the same surface was reused several times (254-255). Please clarify.
- Line 258-260: Figure 2 is a scheme and does not imply you achieved the oriented antibody immobilisation. Therefore, please add relevant data/evidence to back up this claim.
- SPR methods/results: Could the authors explain/clarify if a control was used in performing SPR? Also, you should state for how long you injected the sample and the washing buffer.
- Figure 4 A: the graph seems to show a carry-over effect.
- Line 272-275: please explain how a lower transfer charge resistance can improve the sensor response/assay.
- Line 275: "the foundation of the TBBT SAM on the Au electrode platform is more reproducible in comparison to the 1,6-hexanedithiol dithiol SAM.".Please rephrase and explain better this concept.
- Check your LOD; it seems you are not taking into account the control and blank readings.
- You use BSA as a blocking agent and to test the selectivity. I suspect the sensor surface BSA was already saturated when you blocked the surface using 1mL of PBS 0.1M pH7.4. This might explain why you did not experience any relevant sensor readings. Above all, protein occurring in the real sample should be selected for selectivity study. Please explain why you decided to test the sensor selectivity using Bovine serum albumin.
- Line 374 -376, please add a reference here and explain why it is relevant to know when the collagen synthesis starts in the patient.
Author Response
Reviewer #4.
The manuscript presents an interesting work on the detection of collagen I using the SPR and EIS biosensors platform. Although the manuscript is of interest, some major corrections - listed below- are recommended prior to process the manuscript further. Please do not feel intimidated by the long list, some are actually minor corrections.
Comment 1. The Author should explain better why physicians or patients need to detect/monitoring of collagen. This was presented only partially. It would be good to know how knowing a specific concentration of collagen I can help in the treatment plan and how this biosensor can be beneficial over existing methods.
Response: As suggested, authors have improved and added some information regarding the importance to determine as small as possible detection of the collagen type I content from the biopsy of damaged tendon and ligament tissue in order to improve diagnosis procedure and further type of treatment, which can base either on the direct injection of the doses of hyaluronic acid and collagen or application of necessity of the invasive surgery, Moreover, we included the information regarding the morphological structure and morphology of tendons and ligaments mostly composed form the connective tissue and collagen type I and III with proteoglycans and elastin. However, a low vascularization level has a high impact on the limited capability to regenerate the ligaments. It is demanded to verify even the small presence of the appropriate type of collagen I in the range of picograms to accelerate the selection of the most suitable type of treatment.
‘Due to the fact that the tendons and ligaments exhibit high tensile strength[11], because there are mostly composed of the connective tissue, proteoglycans, elastin and collagen type I and III fibrils with spindle shape tenocytes. [12,13] Nevertheless, this contributes to the low vascularization of those tissues and reduce their capability for efficient regeneration. [14] To improve and accelerate the regeneration mechanism, it is demanded to initially very precisely evaluate the presence of collagen type I, which indicate the appropriate selection of therapy [15]or surgical reconstruction.[16]’
Comment 2. The Authors should explain why the concentration range they reached is advantageous over the concentration ranges achieved by existing test, such as ELISA. In other words, why the concentration in the pg/mL range is better than the concentration range in ng/mL. Is there any clinical/diagnostic benefit?
Response: The authors pointed out this in the introduction section and explained in more detail why the improvement of the more sensitive biomarker on the level of picograms can contribute to developing the current diagnosis system, which is based mostly on the ELISA test with accuracy of the ng/ml level. We explained that the presence of the even much smaller content of collagen type I in the picograms can induce and accelerate the selection of appropriate treatment earlier and apply for example different doses of the hyaluronic acid or collagen type I/III injection against to implementation of the highly invasive surgery.
‘Currently, performed techniques are based on the biopsy of the damaged tendon or ligament tissue and usually allow to determine the collagen content with the maximum accuracy of ng/ml. [17] Accordingly, the application of in-situ direct biomarkers to evaluate the content of the collagen type I [18] with a highly sensitive level in the picomolar range can significantly improve the diagnosis of the occurrence of the potential healing process and allow for the appropriate treatment sooner using either percutaneous injection of collagen and hyaluronic acid or implementation of the invasive surgery[15]. To increase the retreatment process, it is necessary to indicate the concentration of collagen type I with which directly enhances the regeneration of damaged tendon tissue. Therefore, it is vital to implement sensitive systems suitable for detecting collagen type I with an accurate level of selectivity.
Comment 3. The Authors should update some part of the bibliography, e.g. Gelse et al., 2003, MAffuli et al. 2004, Etherington et al., 1981, etc. These references should be replaced with more recent works.
Response: As suggested, authors have updated these references with more recent research:
Miller, E.J. Collagen types: Structure, distribution, and functions. In Collagen: Volume I: Biochemistry; 2018 ISBN9781351079242.
Maffulli, N.; Longo, U.G.; Kadakia, A.; Spiezia, F. Achilles tendinopathy. Foot Ankle Surg. 2020.
Kaushik, B.K.; Singh, L.; Singh, R.; Zhu, G.; Zhang, B.; Wang, Q.; Kumar, S. Detection of Collagen-IV Using Highly Reflective Metal Nanoparticles-Immobilized Photosensitive Optical Fiber-Based MZI Structure. IEEE Trans. Nanobioscience 2020, doi:10.1109/TNB.2020.2998520.
Comment 4. The authors claimed that no separation process was needed after the enzymatic digestion of IgG. Please explain how did you reach this conclusion. I was expecting an enzymatic inactivation step/checks.
Response: Authors followed the protocol described properly by Sharma et al. (Sharma, H.; Mutharasan, R. Half antibody fragments improve biosensor sensitivity without loss of selectivity. Anal. Chem. 2013, doi:10.1021/ac3035426.). It was adapted this technique to reduced the IgG antibody through the disulfide bridges by enzymatic reaction using TECP compound. We extended our explanation regarding the fact that it was not necessary to purify the received fractions after digestion of the antibody because all of the other fragments such as half-Fc and Fab with half-Fab have a function in blocking empty spaces onto the receptor layer.
‘In this approach, the further separation process of the reaction mixture was not necessary, because the other fractions of the half-Fc and Fab with half-Fab antibody fragments contribute in the mechanism of the efficient blocking empty spaces onto the receptor layer.’
Comment 5. Although the authors stated that the half IgG was covalently immobilised on the Au/TBBT/AuNPs, it seems that the immobilisation was achieved via adsorption (lines 142-145). Please clarify.
Response: One of the main principles of our performed research is to modify the gold electrode with the antibody receptor through the covalent bond with the transducer substrate, which was clarified in the methodology section. The activated smooth polycrystalline gold surface was modified by the TBBT compound by the deposition of the solution of TBBT in ethanol for 0.5 hr. This process is based on induction the mechanism to form a covalent bond between the thiol groups –SH with Au surface. Accordingly, the second step was related to covalently linked the deposited colloidal dispersion of gold nanoparticles directly deposited onto the modified previously electrode for 2 hrs. This process allowed to bond of the gold nanoparticles through the thiol group from the other side of TBBT, which was demonstrated in Figure 2. The parallel preparation of the digested half-reduced antibody through the enzymatic reaction enables to bind the disulfide bridges side also through the covalent bond with the form SAM layer of the gold nanoparticles. Therefore, the further deposition of the solution contained the reduced antibody half-IgG for 2 hrs allowed to stable covalently immobilized them. The filling of the empty free spaces and eliminating the unspecific binding was performed by the deposition of the BSA solution for 0.5 hr. Consequently, the application of the deposited drop solution of the reduced half-IgG antibody directly on the gold nanoparticles SAM layer ensured the mechanism of the interaction between AuNPs and half-IgG, and eventually form a stable strong covalent bond. During deposition of the drops solution for each step of modification, the electrode was prevented and covered by an Eppendorf tube to avoid evaporation. The content of the high presence of dissociated ions in PBS solution or either colloidal gold contributes to increasing the ionic strength and successfully imitate the mechanism of creation of covalent antibody immobilization. Therefore, it was not a simple process of the physical absorption onto the electrode surface.
‘In this work, we construct a biosensor composed of the 4,4’-thiobisbenzenethiol (TBBT) SAM enabled to covalently bound of gold nanoparticles (AuNPs). It has been applied for the immobilization of half-antibody fragments via metal nanoparticles using disulfide bridge covalent bonds. The half IgG was derived by the process of the enzymatic digestion using the tris(2-carboxyethyl) phosphine hydrochloride (TECP) [65]’
‘The focus of this work was to verify the optimized stage of the electrode modification and detect the collagen I. Application of the 4,4′-thiobisbenzenethiol (TBBT) compound contained two SH groups that played important role in the covalent deposition on the Au substrate, as well as covalent immobilization of Au NPs.’
‘The immunosensor fabrication consists of the following steps (Figure 2): (i) TBBT SAM deposition on the Au electrode, (ii) covalent deposition of AuNPs, (iii) covalent deposition of the half-antibody fragment, and (iv) filling of empty free spaces and eliminating of unspecific binding by BSA.’
Comment 6. Line 157-158: Did the authors use any means to protect the electrodes from photo-degradation?
Response: Authors specified this description in the methodology section because we have used the black type of the Eppendorf tube to prevent the sample from the potential effect of photo-degradation. However, the used analyte of collagen type I does not reveal very low photo-stability, therefore, any additional technique was not required to eliminate the negative influence of the light exposition.
‘ Then, the electrodes were prevented from air contamination and evaporation of the solutions by covering them with black Eppendorf tubes.’
Comment 7. Ferro-ferricyanides is usually referred to as the redox-active probe (line166).
Response: As suggested, the corrections have been made in the revised manuscript.
‘Electrochemical experiments were performed in the electrolyte composed of 0.1 M PBS (aqueous salts solution with 2.7 mM KCl, 137 mM NaCl, 1.8 mM Na2HPO4, 10 mM KH2PO4, pH 7.4) with the addition of 0.5 mM ferro- and ferricyanides (K3[Fe (CN)6] / K4[Fe (CN)6]; (1:1)) as a redox-active probe .’
Comment 8. On AFM study result section: Is the difference in the roughness statistically significant? If so, add the statistical analysis, please. the quality of the AFM images need to be improved. Furthermore, please add a possible explanation on roughness decreased value observed after TBBT immobilization.
Response: Authors improved the results section of the AFM studies. We conducted the ANOVA posthoc Tukey Test to evaluate the significant difference for the roughness results, however, the roughness did not reveal any significant changes between Au and Au/TBBT of the electrode modifications. The average roughness was in the same range as the standard deviations between those groups, and we corrected this information in the manuscript. However, we have conducted also additional measurements of the thickness and it was noted the presence of a significant difference in case of the thickness between the surface after immobilization of gold nanoparticles (Au/TBBT/AuNPs) and the gold electrode functionalized with TBBT compound (Au/TBBT). Accordingly, we incorporated these average thickness results with standard deviations in the AFM results section.
‘The average roughness parameter for the bare Au surface was found to be 21.3 ± 0.7 nm (Fig. 3A). Obtained roughness 19.4 ± 3.6 nm and thickness 21.05 ± 6.3 nm after the immobilization of TBBT on the Au surface (Fig. 3B) was not significantly changed and it was in the same range in relation to the plain gold surface. After the incorporation of Au nanoparticles, the topography of the TBBT modified surface again changed and attached particles became visible in the corresponding AFM images. The roughness of the modified surfaces changed to 22.6 ± 3.8 nm and the measured thickness significantly increased to 27.6 ± 4.3 nm in comparison to the Au/TBBT sample for AuNPs (Fig. 3C), which is corresponded to similar research performed by Park W. et al. [69] for SAM AuNPs layer.’
Comment 9. Line 224: Authors claims "SPR technique is the most convenient tool to study the interfacial interaction between the analyte (antigen) and the immobilized biomolecules (antibody) in real-time" ->, please explain/comparison with other tools and evidence of this claim. Else please just rephrased with a more objective statement, such as claiming that "the SPR technique is an analytical tool to study the interfacial interaction...in real-time".
Response. As suggested, the corrections have been made in the revised manuscript.
“SPR is a direct, label-free, real-time, measurement of binding kinetics and affinity. It is an optical detection method that utilize the conjugation of prisms that permit biomolecular interactions in real- time. The interaction between bio-molecules is analyzed by determining the change in refractive index in real time. This change in refractive index is obtained from the interaction between the immobilized biomolecule and the analyte. It is the most convenient tool to study the interfacial interaction between the analyte (antigen) and the immobilized biomolecules (antibody) in real-time [72–75]. Therefore, we have applied this technique to confirm the immobilization of half-IgG collagen I antibodies on the transducer surface [76–79]. Upon deposition of the reaction mixture obtained after antibody digestion using TECP [65], an increase of the SPR angle was observed. Real-time, label-free biomolecular interactions between half-IgG and Colla-gen 1 were recorded using Autolab Springle SPR system (Eco Chemie, Netherlands). The 50 nm thick gold coated glass disc was supplied along with the instrument. It is an open cuvette based dual channel system, where channel-1 was used to measure interactions between half-IgG and Collagen 1 and channel-2 was used to monitor signals due to change in refractive index of the buffers and act as reference. Different reagents, samples and buffers were injected in desired amount in two cuvettes (assembled over gold disc). This SPR technique is used to characterize binding interactions be-tween half-IgG and Collagen 1 without any labeling requirements.”
Comment 10. Figure placeholder mismatch and figure missing. I could not locate Figure 2S (line(230). Please double check.
- I can only see Figure 1S and 2S only in the Supplementary info, and these refer to the AFM study.
Response: Authors added and revised the supplementary information with Figure 3S. (corresponded to the curve of SPR confirmation of half-IgG immobilization and we corrected this in the manuscript of the SPr results section.
‘After the deposition of AuNPs on the Au surface the angle shifted from 100 to 1000 and later after immobilization of half-IgG antibodies (Au/TBBT/AuNPs/half-IgG) the angle shifted to 1550 as shown in Figure 3S.’
- The figure placeholder should be consistent between the main text and the figure caption. E.g., Figure 4 (a) in line 242 and figure caption reported as Figure 4 A.
Response: As suggested, the corrections have been made in the revised manuscript.
‘SPR response signal increases consistently upon the exposure of collagen I up to 5 pg/ml, after that response signal decrease as shown in Figure 4A.’
‘. Figure 4B depicts the calibration curve of the SPR signal attained as a function of collagen I concentrations and signifies linearity between 1 pg/ml and 5 pg/ml.’
Comment 11. It is unclear if the Authors developed a regeneration protocol for the SPR sensor as they stated that the same surface was reused several times (254-255). Please clarify.
Response: In this statement, the author meant for the Au/TBBT/AuNPs surface. The developed Au/TBBT/AuNPs layer was placed inside the SPR onto the top of the prism, and then several readings were recorded by changing the interaction spots onto the Au/TBBT/AuNPs surface. SPR interaction readings were taken by immobilizing the antibody on different-different areas/locations of on the same Au/TBBT/AuNPs surface.
Comment 12. Line 258-260: Figure 2 is a scheme and does not imply you achieved the oriented antibody immobilisation. Therefore, please add relevant data/evidence to back up this claim.
Response: The visualization of the scheme in Figure 2 demonstrates the procedure and methodology applied in our research to obtain the highly orientated half-reduced antibody. We have following the protocol published by described properly by Sharma et al. (Sharma, H.; Mutharasan, R. Half antibody fragments improve biosensor sensitivity without loss of selectivity. Anal. Chem. 2013, doi:10.1021/ac3035426.). It was adapted this technique to reduce the IgG antibody through the disulfide bridges by enzymatic reaction using TECP (bond- breaker) compound. This procedure was already optimized in that research regarding applied concertation and incubation time of TECP to efficiently obtain IgG half-reduced antibody, therefore, it was not necessary to evaluate this in the case of our research. The evaluation using the EIS and CV electrochemistry measurements confirmed the successful immobilization of those antibodies after the formation of the SAM AuNPs layer (through the increasing of the electron transfer resistance value from 120 kΩ to 256 kΩ. Moreover, it was checked as well by independent SPR technique, and it was recorded the increasing the SPR angle shift (155⁰) after immobilization of antibody when the sample in between was rinsed by PBS buffer. Therefore, the successful immobilization of half-reduced antibody was confirmed by electrochemical method (CV and EIS) and optical (SPR), which indicate receiving the covalent immobilization of the highly specific and orientated half-IgG antibody as a receptor against to detect collagen I protein.
‘The immobilization of the half-IgG caused a substantial decrease in electrode reversibility. The CV peaks separation increased to 358 ± 76 mV for AuNPs. CV data were also confirmed by EIS. After immobilization of half-IgG, the charge transfer resistance increased to 256 ± 35 kΩ in the case of AuNPs. These parameters confirmed the successful deposition of half-IgG on the nanoparticles.’
‘SPR technique is the most convenient tool to study the interfacial interaction between the analyte (antigen) and the immobilized biomolecules (antibody) in real-time [72–75]. Therefore, we have applied this technique to confirm the im-mobilization of half-IgG collagen I antibodies on the transducer surface [76–79]. Upon deposition of the reaction mixture obtained after antibody digestion using TECP [65], an increase of the SPR angle was observed.
The SPR angle was increased from 100 to 1550 after the immobilization of the antibody on the surface of Au. After the deposition of AuNPs on the Au surface the angle shifted from 100 to 1000 and later after immobilization of half-IgG antibodies (Au/TBBT/AuNPs/half-IgG) the angle shifted to 1550 as shown in Figure 23S. After the immobilization of half-IgG steady-state conditions were obtained, the surface was washed using PBS buffer to remove unbounded species. The obtained results validated the successful half-IgG antibody immobilization on the Au/TBBT/AuNPs surface within approximately 1 hr and 20 minutes (Figure 3S).’
Comment 13. SPR methods/results: Could the authors explain/clarify if a control was used in performing SPR? Also, you should state for how long you injected the sample and the washing buffer.
Response: Control study was performed by exposing Au/TBBT/AuNPs directly to the Collagen 1 but no change observed in the SPR response angle as this technique is based on direct and specific binding interactions between half-IgG and Collagen 1 without any labelling requirements. The antigen-antibody interaction were measured by injecting the Collagen 1 for 6 to 10 min followed by a rinsing period of 15-20 minutes with pure running buffer. The association phase was reached within 10 mins.
Comment 14. Figure 4 A: the graph seems to show a carry-over effect.
Response: Authors are unable to understand this point well. But please be noted that authors optimized all the operational parameter with reference to every experiment reported in this research. In this direction, each curve in Figure 4 A is recorded using the identical optimized conditions.
Comment 15. Line 272-275: please explain how a lower transfer charge resistance can improve the sensor response/assay.
Response: As suggested, authors added some further explanation regarding the fact that lower electron transfer resistance contributes to increase the higher electrical conductivity and enhance the signal. This also plays important role in the detection with a wider higher range of the analyte concentration with the elimination of the potential effect of the electrode blocking.
‘This value is three times lower than the charged transfer resistance of 1,6-hexanedithiol dithiol [52,53]. Accordingly, it enables to receive a higher electrical signal, detecting the analyte with a wider concentration range and reduced the potential negative effect of the electrode blocking. '
Comment 16. Line 275: "the foundation of the TBBT SAM on the Au electrode platform is more reproducible in comparison to the 1,6-hexanedithiol dithiol SAM.". Please, rephrase and explain better this concept.
Response: As suggested authors modified the sentence regarding an explanation of the advantages of the application Au/TBBT modification in comparison of hexa1,6-hexanedithiol such as lower electron transfer resistance has the influence to increase the electrical conductivity, and consequently amplify the received signal.
‘In addition, the foundation of the TBBT SAM on the Au electrode platform is more reproducible and it amplifies the intensity of recorded signal, because of higher electrical conductivity in comparison to the 1,6-hexanedithiol dithiol SAM. Therefore, the TBBT SAM was applied in the present research.’
Comment 17. Check your LOD; it seems you are not taking into account the control and blank readings.
Response: According to the previous comment (18th, #2nd reviewer), authors modified and recalculated the value of LOD value 0.38 pg/ml using this time the obtained standard deviation of the y-intercept (0.025) and slope (0.22) from the linear regression and consequently, it was correct in the manuscript. To calculate the values of the relative change of the electron transfer resistance, we considered the control blank sample (fully modified electrode without any addition of the collagen analyte) for each applied collagen concentration.
‘To obtain appropriate calibration curves, the relative changes of electron transfer resistance (ΔR) are expressed by the following equation [52]:
R0 represents the value for the electron transfer resistance value of the sensing system recovered in the 0.1 M PBS buffer without application of the analyte. The values of relative changes of electron transfer resistance increased proportionally with higher concentrations of collagen I for the studied system (Figure 7). The slope of the calibration curve and the range of standard deviations determined the precision sensing of collagen I.
The limit of detection (LOD) was determined by using the following formula [84]:
where σ is the value of the standard deviation for the y-intercept and S represents a slope of the regression line. The determined value of LOD for collagen immunosensor was 0.38 pg/ml’
Comment 18. You use BSA as a blocking agent and to test the selectivity. I suspect the sensor surface BSA was already saturated when you blocked the surface using 1mL of PBS 0.1M pH7.4. This might explain why you did not experience any relevant sensor readings. Above all, protein occurring in the real sample should be selected for selectivity study. Please explain why you decided to test the sensor selectivity using Bovine serum albumin.
Response: In the present research, authors used the entire suspension of the digested reduced antibody with receptor half reduced antibody half-IgG and as well half fragment Fc and half Fab fragments, which have already played important functions to block free spaces. However, the last modification with BSA allowed eliminating unspecific binding. After that, the fully modified electrode was rinsed with PBS buffer to eliminate excess physically absorbed and unbounded BSA. Even though electrode have been saturated with BSA in the last step, if our sensor would not be selective the antibody receptor could interact and blocked after the addition of a higher concentration of BSA. According to the electrochemical results of the EIS spectra the range of the electron transfer resistance where on a similar level as it was noted for the addition of the collagen I analyte. Consequently, we proofed that the receptor layer hasn’t blocked and it didn’t reveal the tendency of the increasing electron transfer resistance as it was recorded for the verification of the collagen. Moreover, mostly in similar studies for the collagen detection albumin (Sankiewicz, A.; Lukaszewski, Z.; Trojanowska, K.; Gorodkiewicz, E. Determination of collagen type IV by Surface Plasmon Resonance Imaging using a specific biosensor. Anal. Biochem. 2016, doi:10.1016/j.ab.2016.10.002.) as applied in the selectivity studies, because serum albumin is one of the major proteins composed in the blood plasma and it might interfere during the verification of the collagen type I content from the testing sample form the patient.
Comment 19: Line 374 -376, please add a reference here and explain why it is relevant to know when the collagen synthesis starts in the patient.
Response: As suggested, authors improved this explanation in the introduction section and as well in the results to describe the necessity of the verification small range of the presence of collagen type I in the picomolar concentration and potential impact for the quick evaluation and diagnosis process of the capacity to retreatment using either injection contained collagen type I/III and hyaluronic acid or application straight forward required invasive surgery.
' The verification of the collagen type I concentration in the range of the picomolar has a significant impact for the initial and rapid diagnosis of the regeneration mechanism of tendon and ligaments [13,16–18]. Accordingly, it allows selecting the appropriate form of the treatment, while it will be revealed potential capability for healing them supported by the injection with collagen type I/III and hyaluronic acid or applying directly invasive surgery. [10,15]’
Round 2
Reviewer 2 Report
Comments for authors
biosensors-1245836
Title: Impedimetric and Plasmonic sensing of Collagen I using half antibody supported Au modified self-assembled monolayer system
Overview and comments.
The authors have revised and addressed several questions. However, there are still some points that need to be verified. Please, see below.
a. Comment 3. Since Ω.cm is the unit for resistivity, then please correct “low resistance” in the text by “low resistivity” in “Application of the SAM gold nanoparticles reveals high electrical conductivity and low resistance”.
b. Comment 6. I still miss the reason for the choice of EIS as the main technique for target quantification. I think the authors could include a paragraph in the introduction section about the reasons for the choice.
c. Comments 9 and 10. A rule for stating uncertainties is that it should be rounded to one significant figure (Taylor 1997). Please, correct accordingly.
d. Comment 11. Please, show the data discussed for SPR assessment (include SRP plot containing all steps). Also, discuss in SI and inform in Figure 3S each step, so the reader will understand and correlate each shift in the plot with a specific step.
e. Comments 15 and 16. Figure 5 has no change in the last version. In figure 6, please plot using colors to differentiate the concentrations, otherwise is confusing (e.g., concentrations of 1 and 4 pg/mL have been plotted with the same symbols and colors).
f. Comment 17. Sames as comments 9 and 10. Also, please correct the phrase. Is it a “reversible” (as included in the manuscript) or a “quasi-reversible” (as discussed in your response) process?
Reference
Taylor, J.R., 1997. The study of uncertainties in physical measurements, second ed. University Science Books.
Author Response
Response to the Reviewers Comments.
Journal: Biosensors
Manuscript ID: biosensors-1245836
Title: Impedimetric and Plasmonic sensing of Collagen I using half antibody supported Au modified self-assembled monolayer system
Special Issue: Advanced Electrochemical and Opto-Electrochemical Biosensors for Quantitative Analysis of Disease Markers and Viruses
Assigned Editor: Leira Hao
Reviewer #2
Overview and comments.
The authors have revised and addressed several questions. However, there are still some points that need to be verified. Please, see below.
- Comment 3. Since Ω.cm is the unit for resistivity, then please correct “low resistance” in the text by “low resistivity” in “Application of the SAM gold nanoparticles reveals high electrical conductivity and low resistance”.
Response: As suggested, the corrections have been made in revised manuscript.
‘Application of the SAM gold nanoparticles reveals high electrical conductivity and low resistivity’
- Comment 6. I still miss the reason for the choice of EIS as the main technique for target quantification. I think the authors could include a paragraph in the introduction section about the reasons for the choice.
Response: Authors have included this information regarding the choice of the application of EIS for the detection of the collagen type I in the introduction section. We explained the good performance of EIS in relation to the very high specificity as a quantify method allows to determine the specific value of the electron transfer resistance correlated to the corresponding concentration of the analyzed antigen. We added necessary references and explanation that the set parameters of the bias potential and frequency range have lower impact for the stability of the electrode in comparison to the CV technique required wider range of the potential. We included the information of the wide utilization of EIS for the evaluation of the electrode stability, and process of the protein absorption, exchanging of ions, diffusion or rate of the charge transfer.
‘Implementation of the Electrochemical Spectroscopy Impedance as a quantify method enables to record very precisely electrochemical signal of the electron transfer resistance with high sensitivity. [66] The adapted parameters of EIS such as bias potential and frequency ensure have not negative impact for the stability of the receptor layer in comparison to the cyclic voltammetry (CV), which required wide range of the applied potential.[67] Impedance spectroscopy plays important role to evaluate the electrochemical condition, stability of the sensor electrode, and detect the rate of charge transfer, absorption of the proteins, ion exchange and interaction between the antibody-antigen recognition. [68] All of those factors have fundamental aspect to utilize the EIS technique for the highly sensitive and specific biosensor, where determined values of the electron transfer resistance are correlated with the accessibility of the used redox marker to the electrode interface and it allows to verifying very accurate the concentration of the analyte.[69] Subsequently, the Electrochemical Impedance Spectroscopy was selected for the one of the used methods for the performed collagen I sensing platform’c. Comments 9 and 10. A rule for stating uncertainties is that it should be rounded to one significant figure (Taylor 1997). Please, correct accordingly.
Response: Accordingly, we corrected this in the results of thickness for the stating uncertainties rounded to one significant value, in this case till one tenth parts.
‘The summary comparison with the measured values for the average roughness (Ra) and thickness were presented at Figure 2S with marked presence of the statistically significant differences. The average roughness parameter for the bare Au surface was found to be 21.3 ± 0.7 nm (Fig. 3A). Obtained roughness 19.4 ± 3.6 nm and thickness 21.1 ± 6.3 nm after the immobilization of TBBT on the Au surface (Fig. 3B) was not significantly changed and it was in the same range in relation to the plain gold surface. After the incorporation of Au nanoparticles, the topography of the TBBT modified surface again changed and attached particles became visible in the corresponding AFM images. The roughness of the modified surfaces changed to 22.6 ± 3.8 nm and the measured thickness significantly increased to 27.6 ± 4.3 nm in comparison to the Au/TBBT sample for AuNPs (Fig. 3C), which is corresponded to similar research performed by Park W. et al. [73] for SAM AuNPs layer’
- Comment 11. Please, show the data discussed for SPR assessment (include SRP plot containing all steps). Also, discuss in SI and inform in Figure 3S each step, so the reader will understand and correlate each shift in the plot with a specific step.
Response: Figure 4SA shows the change in SPR angle after each step of electrode fabrication.
The Figure 4SA represents the change in the reflectivity angle with each step of electrode fabrication. The study on plasmon dip variation in the process of fabrication of LIP/GO/4-ATP/Au SPR surface reveals useful information on the plasmonic behavior of the SPR surface and the depth of the surface layer. It demonstrates the shift of SPR angle with the change in refractivity percentage for (i) Au, (ii) Au/TBBT (iii) Au/TBBT/AuNPs and (iv) Au/TBBT/AuNPs/half-IgG surfaces. Self-assembly of insulating TBBT on Au SPR surface is confirmed by an angle shift and reduction of plasmon dip of the peak. Thus, fabricated insulating TBBT change the refractive index of the Au/TBBT SPR surface. After the attachment of AuNPs broadening of the peak and reduction of plasmon dip is noticed. Further, a shift of response angle, broading of the peak and decrease in plasmon dip indicates immobilization of insulating half-IgG onto the Au/TBBT/AuNPs surface.
- Comments 15 and 16. Figure 5 has no change in the last version. In figure 6, please plot using colors to differentiate the concentrations, otherwise is confusing (e.g., concentrations of 1 and 4 pg/mL have been plotted with the same symbols and colors).
Response: As suggested Figure 5 and Figure 6 have been changed. We changed the scale for both x and y-axis as orthogonal for all the presented EIS graphs with the same range of scale to eliminate the graph distortions. Moreover, we included the table with fitting results to the Randles electrical circuit for the calculated value for Ret, Rs and CPE with the standard SI units after each step of the electrode modification. We added the different colors and symbols assigned for the different stage of the modification and analyte concentrations.
- Comment 17. Sames as comments 9 and 10. Also, please correct the phrase. Is it a “reversible” (as included in the manuscript) or a “quasi-reversible” (as discussed in your response) process?
Response: As suggested, the corrections have been made in revised manuscript.
‘This caused a decrease in the accessibility of the ferro- and ferricyanides [Fe (CN)6]3−/4− redox marker to the interface with the working electrode surface. Consequently, the oxidation and reduction peaks separation increased to 386 ± 44 mV and the electron transfer resistance (Ret) estimated with EIS was 560 ± 58 kΩ. Both parameters confirmed the successful deposition of TBBT SAM. After the immobilization of the AuNPs, the difference between the oxidation and reduction peak potential decreased to the value 168 ± 18 mV (Figure 5A). Correspondingly, the charge transfer resistance was reduced to 120 ± 11 kΩ (Figure 5B). This result indicated the presence of the quasi-reversible electron transfer mechanism between the transducer and redox marker [87], and subsequently enhanced to form a higher electric conductivity interface similar to the bare gold electrode. ‘
Reference
Taylor, J.R., 1997. The study of uncertainties in physical measurements, second ed. University Science Books.
Reviewer #4
Comments and Suggestions for Authors
Comment 1. The Author should explain better why physicians or patients need to detect/monitoring of collagen. This was presented only partially. It would be good to know how knowing a specific concentration of collagen I can help in the treatment plan and how this biosensor can be beneficial over existing methods.
Response: As suggested, authors have improved and added some information regarding the importance to determine as small as possible detection of the collagen type I content from the biopsy of damaged tendon and ligament tissue in order to improve diagnosis procedure and further type of treatment, which can base either on the direct injection of the doses of hyaluronic acid and collagen or application of necessity of the invasive surgery, Moreover, we included the information regarding the morphological structure and morphology of tendons and ligaments mostly composed form the connective tissue and collagen type I and III with proteoglycans and elastin. However, a low vascularization level has a high impact on the limited capability to regenerate the ligaments. It is demanded to verify even the small presence of the appropriate type of collagen I in the range of picograms to accelerate the selection of the most suitable type of treatment.
‘Due to the fact that the tendons and ligaments exhibit high tensile strength[11], because there are mostly composed of the connective tissue, proteoglycans, elastin and collagen type I and III fibrils with spindle shape tenocytes. [12,13] Nevertheless, this contributes to the low vascularization of those tissues and reduce their capability for efficient regeneration. [14] To improve and accelerate the regeneration mechanism, it is demanded to initially very precisely evaluate the presence of collagen type I, which indicate the appropriate selection of therapy [15] or surgical reconstruction.[16]’
Reviewer’s New Comment: Lines 59 - 64, please rephrase as it is not clearly written. Furthermore, I would suggest including your response in the manuscript as it is clearer "the presence of the even much smaller content of collagen type I in the picograms can induce and accelerate the selection of appropriate treatment earlier and apply for example different doses of the hyaluronic acid or collagen type I/III injection against to implementation of the highly invasive surgery.". Please also add reference(s) to sustain your statements.
Response: As suggested the authors have corrected and edited, and rephased the paragraph according to the above commend and we added that statement of the importance of the application much more sensitive collagen sensors and their role in the implementation appropriate doses of the hyaluronic acid injection with appropriate references.
‘Additionally, the determination of collagen type I protein has an important function in the case of tendon inflammation [8–10]. Due to the fact that the tendons and ligaments exhibit high tensile strength[11], because there are mostly composed of the connective tissue, proteoglycans, elastin and collagen type I and III fibrils with spindle shape tenocytes. [12,13] Nevertheless, this structure contributes to the low vascularization of those tissues and reduce their capability for efficient regeneration. [14] To improve and accelerate the regeneration mechanism, it is demanded to initially very precisely evaluate the presence of collagen type I,[15] which indicate the occurrence mechanism of synthesis of the collagenous fibrils and starting self-healing process. [10] This directly determines the appropriate selection of applied therapies [16]or invasive surgical reconstruction.[15] Currently, performed techniques are based on the biopsy of the damaged tendon or ligament tissue and usually allow to determine the collagen content with the maximum accuracy of ng/ml. [17] Accordingly, the application of in-situ direct biomarkers to evaluate the content of the collagen type I [18] with a highly sensitive level in the picomolar range can significantly improve the diagnosis of the occurrence of the potential healing process and allow for the appropriate treatment sooner using either percutaneous injection of collagen and hyaluronic acid or implementation of the invasive surgery [16]. The presence of the even much smaller content of collagen type I in the picograms can induce and accelerate the selection of appropriate treatment earlier and apply for example different doses of the hyaluronic acid or collagen type I/III injection against to implementation of the highly invasive surgery. [10,15,16] To increase the retreatment process, it is significantly relevant to indicate the concentration of collagen type I which directly enhances the regeneration of damaged tendon tissue. For this reason, it is required to implement sensitive systems suitable for detecting collagen type I with an accurate level of selectivity.’
Comment 2. The Authors should explain why the concentration range they reached is advantageous over the concentration ranges achieved by existing test, such as ELISA. In other words, why the concentration in the pg/mL range is better than the concentration range in ng/mL. Is there any clinical/diagnostic benefit?
Response: The authors pointed out this in the introduction section and explained in more detail why the improvement of the more sensitive biomarker on the level of picograms can contribute to developing the current diagnosis system, which is based mostly on the ELISA test with accuracy of the ng/ml level. We explained that the presence of the even much smaller content of collagen type I in the picograms can induce and accelerate the selection of appropriate treatment earlier and apply for example different doses of the hyaluronic acid or collagen type I/III injection against to implementation of the highly invasive surgery.
‘Currently, performed techniques are based on the biopsy of the damaged tendon or ligament tissue and usually allow to determine the collagen content with the maximum accuracy of ng/ml. [17] Accordingly, the application of in-situ direct biomarkers to evaluate the content of the collagen type I [18] with a highly sensitive level in the picomolar range can significantly improve the diagnosis of the occurrence of the potential healing process and allow for the appropriate treatment sooner using either percutaneous injection of collagen and hyaluronic acid or implementation of the invasive surgery[15]. To increase the retreatment process, it is necessary to indicate the concentration of collagen type I with which directly enhances the regeneration of damaged tendon tissue. Therefore, it is vital to implement sensitive systems suitable for detecting collagen type I with an accurate level of selectivity.
Reviewer’s New Comment: thanks now it is clearer.
Comment 3. The Authors should update some part of the bibliography, e.g. Gelse et al., 2003, MAffuli et al. 2004, Etherington et al., 1981, etc. These references should be replaced with more recent works.
Response: As suggested, authors have updated these references with more recent research:
Miller, E.J. Collagen types: Structure, distribution, and functions. In Collagen: Volume I: Biochemistry; 2018 ISBN9781351079242.
Maffulli, N.; Longo, U.G.; Kadakia, A.; Spiezia, F. Achilles tendinopathy. Foot Ankle Surg. 2020.
Kaushik, B.K.; Singh, L.; Singh, R.; Zhu, G.; Zhang, B.; Wang, Q.; Kumar, S. Detection of Collagen-IV Using Highly Reflective Metal Nanoparticles-Immobilized Photosensitive Optical Fiber-Based MZI Structure. IEEE Trans. Nanobioscience 2020, doi:10.1109/TNB.2020.2998520.
Comment 4. The authors claimed that no separation process was needed after the enzymatic digestion of IgG. Please explain how did you reach this conclusion. I was expecting an enzymatic inactivation step/checks.
Response: Authors followed the protocol described properly by Sharma et al. (Sharma, H.; Mutharasan, R. Half antibody fragments improve biosensor sensitivity without loss of selectivity. Anal. Chem. 2013, doi:10.1021/ac3035426.). It was adapted this technique to reduced the IgG antibody through the disulfide bridges by enzymatic reaction using TECP compound. We extended our explanation regarding the fact that it was not necessary to purify the received fractions after digestion of the antibody because all of the other fragments such as half-Fc and Fab with half-Fab have a function in blocking empty spaces onto the receptor layer.
‘In this approach, the further separation process of the reaction mixture was not necessary, because the other fractions of the half-Fc and Fab with half-Fab antibody fragments contribute in the mechanism of the efficient blocking empty spaces onto the receptor layer.’
Reviewer’s New Comment: To make this claim you need to carry out an experiment with and without separation process and demonstrate that there is no difference.
Also, Sharma et al. used MALDI-MS to demonstrate the success of fab fragmentation. As you are reproducing Sharma et al.'s protocol, you should have carried out this study.
Response: Authors were directly following the same protocol for the enzymatic reduction of the half-IgG antibody using TECP. We used the same type of the whole collagen type I antibody IgG. We assumed that the mechanism of the reducing should analogically occurred as research conducted by Sharma et al. Our laboratory has not got access to perform MALDI-MS. This was not the main aim of our research, TECP bon-breaker is commonly used market product, which are widely used for the reduction of IgG antibodies.
Comment 5. Although the authors stated that the half IgG was covalently immobilized on the Au/TBBT/AuNPs, it seems that the immobilization was achieved via adsorption (lines 142-145). Please clarify.
Response: One of the main principles of our performed research is to modify the gold electrode with the antibody receptor through the covalent bond with the transducer substrate, which was clarified in the methodology section. The activated smooth polycrystalline gold surface was modified by the TBBT compound by the deposition of the solution of TBBT in ethanol for 0.5 hr. This process is based on induction the mechanism to form a covalent bond between the thiol groups –SH with Au surface. Accordingly, the second step was related to covalently linked the deposited colloidal dispersion of gold nanoparticles directly deposited onto the modified previously electrode for 2 hrs. This process allowed to bond of the gold nanoparticles through the thiol group from the other side of TBBT, which was demonstrated in Figure 2. The parallel preparation of the digested half-reduced antibody through the enzymatic reaction enables to bind the disulfide bridges side also through the covalent bond with the form SAM layer of the gold nanoparticles. Therefore, the further deposition of the solution contained the reduced antibody half-IgG for 2 hrs allowed to stable covalently immobilized them. The filling of the empty free spaces and eliminating the unspecific binding was performed by the deposition of the BSA solution for 0.5 hr. Consequently, the application of the deposited drop solution of the reduced half-IgG antibody directly on the gold nanoparticles SAM layer ensured the mechanism of the interaction between AuNPs and half-IgG, and eventually form a stable strong covalent bond. During deposition of the drops solution for each step of modification, the electrode was prevented and covered by an Eppendorf tube to avoid evaporation. The content of the high presence of dissociated ions in PBS solution or either colloidal gold contributes to increasing the ionic strength and successfully imitate the mechanism of creation of covalent antibody immobilization. Therefore, it was not a simple process of the physical absorption onto the electrode surface.
‘In this work, we construct a biosensor composed of the 4,4’-thiobisbenzenethiol (TBBT) SAM enabled to covalently bound of gold nanoparticles (AuNPs). It has been applied for the immobilization of half-antibody fragments via metal nanoparticles using disulfide bridge covalent bonds. The half IgG was derived by the process of the enzymatic digestion using the tris(2-carboxyethyl) phosphine hydrochloride (TECP) [65]’
‘The focus of this work was to verify the optimized stage of the electrode modification and detect the collagen I. Application of the 4,4′-thiobisbenzenethiol (TBBT) compound contained two SH groups that played important role in the covalent deposition on the Au substrate, as well as covalent immobilization of Au NPs.’
‘The immunosensor fabrication consists of the following steps (Figure 2): (i) TBBT SAM deposition on the Au electrode, (ii) covalent deposition of AuNPs, (iii) covalent deposition of the half-antibody fragment, and (iv) filling of empty free spaces and eliminating of unspecific binding by BSA.’
Reviewer’s New Comment: Once again Figure 2 only display the chemistry behind your attachment, and does not include any results which demonstrate you have formed a thiol mediated covalent bond. Please use appropriate words.
Response: Figure 1 and Figure 2 are only schematic representation of the applied methodology. In case of the enzymatic reduction of the IgG collagen type I antibody the evaluation of the efficiency was not necessary regarding to the fact application commonly used enzymatic bond breaker TECP, we followed published earlier by Sharma et al. protocol. However, according to the Figure 2. The particular stage were confirmed electrochemically using CV and EIs (figure 5). Moreover, formation of the AuNPs were confirmed by the AFM topography were observed increasing roughness and thickness. The immobilization of antibody additionally were evaluated by optical technique using SPR. As it was mentioned in the methodology section after each step of the modification electrodes were rinsed several times by the PBS solution, we have worked in the pH 7.4, the electrostatic immobilization or physical absorption could not occur after rinsing electrode by electrode in this pH condition. Therefore, the EIS, CV, AFM and SPR study confirmed the covalent bond between TBBT and Au surface, TBBT and Au NPs, and also AuNPs and half-IgG.
Comment 6. Line 157-158: Did the authors use any means to protect the electrodes from photo-degradation?
Response: Authors specified this description in the methodology section because we have used the black type of the Eppendorf tube to prevent the sample from the potential effect of photo-degradation. However, the used analyte of collagen type I does not reveal very low photo-stability, therefore, any additional technique was not required to eliminate the negative influence of the light exposition.
‘ Then, the electrodes were prevented from air contamination and evaporation of the solutions by covering them with black Eppendorf tubes.’
Comment 7. Ferro-ferricyanides is usually referred to as the redox-active probe (line166).
Response: As suggested, the corrections have been made in the revised manuscript.
‘Electrochemical experiments were performed in the electrolyte composed of 0.1 M PBS (aqueous salts solution with 2.7 mM KCl, 137 mM NaCl, 1.8 mM Na2HPO4, 10 mM KH2PO4, pH 7.4) with the addition of 0.5 mM ferro- and ferricyanides (K3[Fe (CN)6] / K4[Fe (CN)6]; (1:1)) as a redox-active probe .’
Comment 8. On AFM study result section: Is the difference in the roughness statistically significant? If so, add the statistical analysis, please. the quality of the AFM images need to be improved. Furthermore, please add a possible explanation on roughness decreased value observed after TBBT immobilization.
Response: Authors improved the results section of the AFM studies. We conducted the ANOVA posthoc Tukey Test to evaluate the significant difference for the roughness results, however, the roughness did not reveal any significant changes between Au and Au/TBBT of the electrode modifications. The average roughness was in the same range as the standard deviations between those groups, and we corrected this information in the manuscript. However, we have conducted also additional measurements of the thickness and it was noted the presence of a significant difference in case of the thickness between the surface after immobilization of gold nanoparticles (Au/TBBT/AuNPs) and the gold electrode functionalized with TBBT compound (Au/TBBT). Accordingly, we incorporated these average thickness results with standard deviations in the AFM results section.
‘The average roughness parameter for the bare Au surface was found to be 21.3 ± 0.7 nm (Fig. 3A). Obtained roughness 19.4 ± 3.6 nm and thickness 21.05 ± 6.3 nm after the immobilization of TBBT on the Au surface (Fig. 3B) was not significantly changed and it was in the same range in relation to the plain gold surface. After the incorporation of Au nanoparticles, the topography of the TBBT modified surface again changed and attached particles became visible in the corresponding AFM images. The roughness of the modified surfaces changed to 22.6 ± 3.8 nm and the measured thickness significantly increased to 27.6 ± 4.3 nm in comparison to the Au/TBBT sample for AuNPs (Fig. 3C), which is corresponded to similar research performed by Park W. et al. [69] for SAM AuNPs layer.’
Reviewer’s New Comment: you claim "the measured thickness significantly increased to 27.6 ± 4.3 nm in comparison to the Au/TBBT sample for AuNPs". Please add the statistical analysis results. The quality of the AFM 3D images needs to be improved. Presently they can not be accepted for publication.
Response: We have added this results in the supplementary data for the conducted one way ANOVA post-hoc Tukey test of the statistical differences between the measured values for the thickness and average roughness. We mentioned about this calculation procedure in the methodology part. We changed the resolution for the Figure 3.with visible scales and units.
‘It was measured the average roughness (Ra) and thickness from each modification stage. The significant statistical differences (at p<0.005) was evaluated by the one-way ANOVA post-hoc Tukey TEST.’
Comment 9. Line 224: Authors claims "SPR technique is the most convenient tool to study the interfacial interaction between the analyte (antigen) and the immobilized biomolecules (antibody) in real-time" ->, please explain/comparison with other tools and evidence of this claim. Else please just rephrased with a more objective statement, such as claiming that "the SPR technique is an analytical tool to study the interfacial interaction...in real-time".
Response. As suggested, the corrections have been made in the revised manuscript.
“SPR is a direct, label-free, real-time, measurement of binding kinetics and affinity. It is an optical detection method that utilize the conjugation of prisms that permit biomolecular interactions in real- time. The interaction between bio-molecules is analyzed by determining the change in refractive index in real time. This change in refractive index is obtained from the interaction between the immobilized biomolecule and the analyte. It is the most convenient tool to study the interfacial interaction between the analyte (antigen) and the immobilized biomolecules (antibody) in real-time [72–75]. Therefore, we have applied this technique to confirm the immobilization of half-IgG collagen I antibodies on the transducer surface [76–79]. Upon deposition of the reaction mixture obtained after antibody digestion using TECP [65], an increase of the SPR angle was observed. Real-time, label-free biomolecular interactions between half-IgG and Colla-gen 1 were recorded using Autolab Springle SPR system (Eco Chemie, Netherlands). The 50 nm thick gold coated glass disc was supplied along with the instrument. It is an open cuvette based dual channel system, where channel-1 was used to measure interactions between half-IgG and Collagen 1 and channel-2 was used to monitor signals due to change in refractive index of the buffers and act as reference. Different reagents, samples and buffers were injected in desired amount in two cuvettes (assembled over gold disc). This SPR technique is used to characterize binding interactions be-tween half-IgG and Collagen 1 without any labeling requirements.”
Comment 10. Figure placeholder mismatch and figure missing. I could not locate Figure 2S (line(230). Please double check.
- I can only see Figure 1S and 2S only in the Supplementary info, and these refer to the AFM study.
Response: Authors added and revised the supplementary information with Figure 3S. (corresponded to the curve of SPR confirmation of half-IgG immobilization and we corrected this in the manuscript of the SPr results section.
‘After the deposition of AuNPs on the Au surface the angle shifted from 100 to 1000 and later after immobilization of half-IgG antibodies (Au/TBBT/AuNPs/half-IgG) the angle shifted to 1550 as shown in Figure 3S.’
- The figure placeholder should be consistent between the main text and the figure caption. E.g., Figure 4 (a) in line 242 and figure caption reported as Figure 4 A.
Response: As suggested, the corrections have been made in the revised manuscript.
‘SPR response signal increases consistently upon the exposure of collagen I up to 5 pg/ml, after that response signal decrease as shown in Figure 4A.’
‘. Figure 4B depicts the calibration curve of the SPR signal attained as a function of collagen I concentrations and signifies linearity between 1 pg/ml and 5 pg/ml.’
Reviewer’s New Comment: Figure 4A -> please rescale x-axis and y-axis. On the y-axis, there is no need to go down to -280.
Response: Figure 4A has been changed as per suggestions.
You claimed: "The antigen-antibody interaction was measured by injecting the Collagen 1 for 6 to 10 min followed by a rinsing period of 15-20 minutes with pure running buffer.". However, Figure 4A shows that the injection time was much longer than the washing time. Could you please clarify? I suggested indicating the start and the end of collagen I injection on the graph (Figure 4 A). Also, the injection can not be expressed as a range of time if you optimised the procedure (as you stated below.
Response: Yes reviewer is right, the statement the wrong. The antigen-antibody interaction was measured by injecting the Collagen 1 for 20 to 25 min followed by a rinsing period of 10 minutes with pure running buffer." “a” point is the start of association phase and “b” represents the dissociation phase and “c” is the buffer phase.
Figure 4B-> you performed 5 injections, but you have 6 points in your calibration curve. Please clarify.
Response: In this response study we injected the Collagen I from 1 to 6 pg/ml but we observed that the after 5 pg/ml the SPR response angle decreased. The last injection i.e. 6 pg/ml is not in the linear range of the calibration curve that means our system is able to detect the Collagen I from 1 pg.ml to 5 pg/ml.
Figure 3S-> This figure does not provide confirmation of IgG immobilization. Please provide the SPR sensor response to each sep of the sensor surface functionalization/modification, i.e., from bare surface Au to surface blocking with BSA, showing the buffer washing between each injection. Please and once again, scale the graph x-axis and y-axis to enable better visualization.
Response: This figure is in cooperated in the supplementary file as Figure 4S A
Comment 11. It is unclear if the Authors developed a regeneration protocol for the SPR sensor as they stated that the same surface was reused several times (254-255). Please clarify.
Response: In this statement, the author meant for the Au/TBBT/AuNPs surface. The developed Au/TBBT/AuNPs layer was placed inside the SPR onto the top of the prism, and then several readings were recorded by changing the interaction spots onto the Au/TBBT/AuNPs surface. SPR interaction readings were taken by immobilizing the antibody on different-different areas/locations of on the same Au/TBBT/AuNPs surface.
Reviewer’s New Comment: This response is quite creative and not acceptable.
Response: We apologies for the unclear statement. The author means that response readings of antibody-antigen interaction can be taken on the same Au/TBBT/AuNPs disc by immobilizing the Antibodies on different-different spots i.e., spot 1, 2, 3, 4 and 5 via SPR (as shown in figure below). We
need not to fabricate the Au/TBBT/AuNPs surface after one reading.
Comment 12. Line 258-260: Figure 2 is a scheme and does not imply you achieved the oriented antibody immobilisation. Therefore, please add relevant data/evidence to back up this claim.
Response: The visualization of the scheme in Figure 2 demonstrates the procedure and methodology applied in our research to obtain the highly orientated half-reduced antibody. We have following the protocol published by described properly by Sharma et al. (Sharma, H.; Mutharasan, R. Half antibody fragments improve biosensor sensitivity without loss of selectivity. Anal. Chem. 2013, doi:10.1021/ac3035426.). It was adapted this technique to reduce the IgG antibody through the disulfide bridges by enzymatic reaction using TECP (bond- breaker) compound. This procedure was already optimized in that research regarding applied concertation and incubation time of TECP to efficiently obtain IgG half-reduced antibody, therefore, it was not necessary to evaluate this in the case of our research. The evaluation using the EIS and CV electrochemistry measurements confirmed the successful immobilization of those antibodies after the formation of the SAM AuNPs layer (through the increasing of the electron transfer resistance value from 120 kΩ to 256 kΩ. Moreover, it was checked as well by independent SPR technique, and it was recorded the increasing the SPR angle shift (155⁰) after immobilization of antibody when the sample in between was rinsed by PBS buffer. Therefore, the successful immobilization of half-reduced antibody was confirmed by electrochemical method (CV and EIS) and optical (SPR), which indicate receiving the covalent immobilization of the highly specific and orientated half-IgG antibody as a receptor against to detect collagen I protein.
‘The immobilization of the half-IgG caused a substantial decrease in electrode reversibility. The CV peaks separation increased to 358 ± 76 mV for AuNPs. CV data were also confirmed by EIS. After immobilization of half-IgG, the charge transfer resistance increased to 256 ± 35 kΩ in the case of AuNPs. These parameters confirmed the successful deposition of half-IgG on the nanoparticles.’
‘SPR technique is the most convenient tool to study the interfacial interaction between the analyte (antigen) and the immobilized biomolecules (antibody) in real-time [72–75]. Therefore, we have applied this technique to confirm the im-mobilization of half-IgG collagen I antibodies on the transducer surface [76–79]. Upon deposition of the reaction mixture obtained after antibody digestion using TECP [65], an increase of the SPR angle was observed.
The SPR angle was increased from 100 to 1550 after the immobilization of the antibody on the surface of Au. After the deposition of AuNPs on the Au surface the angle shifted from 100 to 1000 and later after immobilization of half-IgG antibodies (Au/TBBT/AuNPs/half-IgG) the angle shifted to 1550 as shown in Figure 23S. After the immobilization of half-IgG steady-state conditions were obtained, the surface was washed using PBS buffer to remove unbounded species. The obtained results validated the successful half-IgG antibody immobilization on the Au/TBBT/AuNPs surface within approximately 1 hr and 20 minutes (Figure 3S).’
Reviewer’s New Comment: I can only see schematization in figure 1 and 2 .... Demonstration requires experimental data which can be plotted in an appropriate plot.
Response: Figure 1 and Figure 2 are only schematic representation of the applied methodology. In case of the enzymatic reduction of the IgG collagen type I antibody the evaluation of the efficiency was not necessary regarding to the fact application commonly used enzymatic bond breaker TECP, we followed published earlier by Shrama et al. protocol. However, according to the Figure 2. The particular stage was confirmed electrochemically using CV and EIs (figure 5). Moreover, formation of the AuNPs were confirmed by the AFM topography were observed increasing roughness and thickness. The immobilization of antibody additionally were evaluated by optical technique using SPR. As it was mentioned in the methodology section after each step of the modification electrodes were rinsed several times by the PBS solution, we have worked in the pH 7.4, the electrostatic immobilization or physical absorption could not occur after rinsing electrode by electrode in this pH condition. Therefore, the EIS, CV, AFM and SPR study confirmed the covalent bond between TBBT and Au surface, TBBT and Au NPs, and also AuNPs and half-IgG.
Comment 13. SPR methods/results: Could the authors explain/clarify if a control was used in performing SPR? Also, you should state for how long you injected the sample and the washing buffer.
Response: Control study was performed by exposing Au/TBBT/AuNPs directly to the Collagen 1 but no change observed in the SPR response angle as this technique is based on direct and specific binding interactions between half-IgG and Collagen 1 without any labelling requirements. The antigen-antibody interaction was measured by injecting the Collagen 1 for 6 to 10 min followed by a rinsing period of 15-20 minutes with pure running buffer. The association phase was reached within 10 mins.
Reviewer’s New Comment: Any negative control?
Response: The authors checked the SPR response angle without immobilizing the antibodies onto the Au/TBBT/AuNPs. No change in the angle was observed. We were unable to get any association phase without antibody.
Comment 14. Figure 4 A: the graph seems to show a carry-over effect.
Response: Authors are unable to understand this point well. But please be noted that authors optimized all the operational parameter with reference to every experiment reported in this research. In this direction, each curve in Figure 4 A is recorded using the identical optimized conditions.
Reviewer’s New Comment: Please improve figure 4A as suggested above.
Response: The figure has been improved as per suggestions.
Comment 15. Line 272-275: please explain how a lower transfer charge resistance can improve the sensor response/assay.
Response: As suggested, authors added some further explanation regarding the fact that lower electron transfer resistance contributes to increase the higher electrical conductivity and enhance the signal. This also plays important role in the detection with a wider higher range of the analyte concentration with the elimination of the potential effect of the electrode blocking.
‘This value is three times lower than the charged transfer resistance of 1,6-hexanedithiol dithiol [52,53]. Accordingly, it enables to receive a higher electrical signal, detecting the analyte with a wider concentration range and reduced the potential negative effect of the electrode blocking. '
Comment 16. Line 275: "the foundation of the TBBT SAM on the Au electrode platform is more reproducible in comparison to the 1,6-hexanedithiol dithiol SAM.". Please, rephrase and explain better this concept.
Response: As suggested authors modified the sentence regarding an explanation of the advantages of the application Au/TBBT modification in comparison of hexa1,6-hexanedithiol such as lower electron transfer resistance has the influence to increase the electrical conductivity, and consequently amplify the received signal.
‘In addition, the foundation of the TBBT SAM on the Au electrode platform is more reproducible and it amplifies the intensity of recorded signal, because of higher electrical conductivity in comparison to the 1,6-hexanedithiol dithiol SAM. Therefore, the TBBT SAM was applied in the present research.’
Comment 17. Check your LOD; it seems you are not taking into account the control and blank readings.
Response: According to the previous comment (18th, #2nd reviewer), authors modified and recalculated the value of LOD value 0.38 pg/ml using this time the obtained standard deviation of the y-intercept (0.025) and slope (0.22) from the linear regression and consequently, it was correct in the manuscript. To calculate the values of the relative change of the electron transfer resistance, we considered the control blank sample (fully modified electrode without any addition of the collagen analyte) for each applied collagen concentration.
‘To obtain appropriate calibration curves, the relative changes of electron transfer resistance (ΔR) are expressed by the following equation [52]:
R0 represents the value for the electron transfer resistance value of the sensing system recovered in the 0.1 M PBS buffer without application of the analyte. The values of relative changes of electron transfer resistance increased proportionally with higher concentrations of collagen I for the studied system (Figure 7). The slope of the calibration curve and the range of standard deviations determined the precision sensing of collagen I.
The limit of detection (LOD) was determined by using the following formula [84]:
where σ is the value of the standard deviation for the y-intercept and S represents a slope of the regression line. The determined value of LOD for collagen immunosensor was 0.38 pg/ml’
Comment 18. You use BSA as a blocking agent and to test the selectivity. I suspect the sensor surface BSA was already saturated when you blocked the surface using 1mL of PBS 0.1M pH7.4. This might explain why you did not experience any relevant sensor readings. Above all, protein occurring in the real sample should be selected for selectivity study. Please explain why you decided to test the sensor selectivity using Bovine serum albumin.
Response: In the present research, authors used the entire suspension of the digested reduced antibody with receptor half reduced antibody half-IgG and as well half fragment Fc and half Fab fragments, which have already played important functions to block free spaces. However, the last modification with BSA allowed eliminating unspecific binding. After that, the fully modified electrode was rinsed with PBS buffer to eliminate excess physically absorbed and unbounded BSA. Even though electrode have been saturated with BSA in the last step, if our sensor would not be selective the antibody receptor could interact and blocked after the addition of a higher concentration of BSA. According to the electrochemical results of the EIS spectra the range of the electron transfer resistance where on a similar level as it was noted for the addition of the collagen I analyte. Consequently, we proofed that the receptor layer has not blocked, and it didn’t reveal the tendency of the increasing electron transfer resistance as it was recorded for the verification of the collagen. Moreover, mostly in similar studies for the collagen detection albumin (Sankiewicz, A.; Lukaszewski, Z.; Trojanowska, K.; Gorodkiewicz, E. Determination of collagen type IV by Surface Plasmon Resonance Imaging using a specific biosensor. Anal. Biochem. 2016, doi:10.1016/j.ab.2016.10.002.) as applied in the selectivity studies, because serum albumin is one of the major proteins composed in the blood plasma and it might interfere during the verification of the collagen type I content from the testing sample form the patient.
Reviewer’s New Comment: If you saturated the sensor surface with BSA to achieve the surface blocking, it is obvious you won't achieve any BSA interaction upon BSA further exposure. As it stands, you did not perform the selectivity study.
Response: Even though the electrode were for the step fully modified and saturated using BSA to blocking unspecific binding, then electrode was rinsed several times using PBS to remove any residual unbounded BSA protein. However, further application of longer exposition of BSA did not present any interaction with collagen type I antibody in the condition of PBS electrolyte and redox marker. Bovine Albumin Serum is usually applied as a negative control study in other research regarding the evaluation of the collagen type I concentration (we added appropriate references: (Sankiewicz, A.; Lukaszewski, Z.; Trojanowska, K.; Gorodkiewicz, E. Determination of collagen type IV by Surface Plasmon Resonance Imaging using a specific biosensor. Anal. Biochem. 2016, doi:10.1016/j.ab.2016.10.002.)
Comment 19: Line 374 -376, please add a reference here and explain why it is relevant to know when the collagen synthesis starts in the patient.
Response: As suggested, authors improved this explanation in the introduction section and as well in the results to describe the necessity of the verification small range of the presence of collagen type I in the picomolar concentration and potential impact for the quick evaluation and diagnosis process of the capacity to retreatment using either injection contained collagen type I/III and hyaluronic acid or application straight forward required invasive surgery.
' The verification of the collagen type I concentration in the range of the picomolar has a significant impact for the initial and rapid diagnosis of the regeneration mechanism of tendon and ligaments [13,16–18]. Accordingly, it allows selecting the appropriate form of the treatment, while it will be revealed potential capability for healing them supported by the injection with collagen type I/III and hyaluronic acid or applying directly invasive surgery. [10,15]’

Reviewer 4 Report
Comment 1. The Author should explain better why physicians or patients need to detect/monitoring of collagen. This was presented only partially. It would be good to know how knowing a specific concentration of collagen I can help in the treatment plan and how this biosensor can be beneficial over existing methods.
Response: As suggested, authors have improved and added some information regarding the importance to determine as small as possible detection of the collagen type I content from the biopsy of damaged tendon and ligament tissue in order to improve diagnosis procedure and further type of treatment, which can base either on the direct injection of the doses of hyaluronic acid and collagen or application of necessity of the invasive surgery, Moreover, we included the information regarding the morphological structure and morphology of tendons and ligaments mostly composed form the connective tissue and collagen type I and III with proteoglycans and elastin. However, a low vascularization level has a high impact on the limited capability to regenerate the ligaments. It is demanded to verify even the small presence of the appropriate type of collagen I in the range of picograms to accelerate the selection of the most suitable type of treatment.
‘Due to the fact that the tendons and ligaments exhibit high tensile strength[11], because there are mostly composed of the connective tissue, proteoglycans, elastin and collagen type I and III fibrils with spindle shape tenocytes. [12,13] Nevertheless, this contributes to the low vascularization of those tissues and reduce their capability for efficient regeneration. [14] To improve and accelerate the regeneration mechanism, it is demanded to initially very precisely evaluate the presence of collagen type I, which indicate the appropriate selection of therapy [15]or surgical reconstruction.[16]’
Response: Lines 59 - 64, please rephrase as it is not clearly written. Furthermore, I would suggest including your response in the manuscript as it is clearer "the presence of the even much smaller content of collagen type I in the picograms can induce and accelerate the selection of appropriate treatment earlier and apply for example different doses of the hyaluronic acid or collagen type I/III injection against to implementation of the highly invasive surgery.". Please also add reference(s) to sustain your statements.
Comment 2. The Authors should explain why the concentration range they reached is advantageous over the concentration ranges achieved by existing test, such as ELISA. In other words, why the concentration in the pg/mL range is better than the concentration range in ng/mL. Is there any clinical/diagnostic benefit?
Response: The authors pointed out this in the introduction section and explained in more detail why the improvement of the more sensitive biomarker on the level of picograms can contribute to developing the current diagnosis system, which is based mostly on the ELISA test with accuracy of the ng/ml level. We explained that the presence of the even much smaller content of collagen type I in the picograms can induce and accelerate the selection of appropriate treatment earlier and apply for example different doses of the hyaluronic acid or collagen type I/III injection against to implementation of the highly invasive surgery.
‘Currently, performed techniques are based on the biopsy of the damaged tendon or ligament tissue and usually allow to determine the collagen content with the maximum accuracy of ng/ml. [17] Accordingly, the application of in-situ direct biomarkers to evaluate the content of the collagen type I [18] with a highly sensitive level in the picomolar range can significantly improve the diagnosis of the occurrence of the potential healing process and allow for the appropriate treatment sooner using either percutaneous injection of collagen and hyaluronic acid or implementation of the invasive surgery[15]. To increase the retreatment process, it is necessary to indicate the concentration of collagen type I with which directly enhances the regeneration of damaged tendon tissue. Therefore, it is vital to implement sensitive systems suitable for detecting collagen type I with an accurate level of selectivity.
Response: thanks now it is clearer.
Comment 3. The Authors should update some part of the bibliography, e.g. Gelse et al., 2003, MAffuli et al. 2004, Etherington et al., 1981, etc. These references should be replaced with more recent works.
Response: As suggested, authors have updated these references with more recent research:
Miller, E.J. Collagen types: Structure, distribution, and functions. In Collagen: Volume I: Biochemistry; 2018 ISBN9781351079242.
Maffulli, N.; Longo, U.G.; Kadakia, A.; Spiezia, F. Achilles tendinopathy. Foot Ankle Surg. 2020.
Kaushik, B.K.; Singh, L.; Singh, R.; Zhu, G.; Zhang, B.; Wang, Q.; Kumar, S. Detection of Collagen-IV Using Highly Reflective Metal Nanoparticles-Immobilized Photosensitive Optical Fiber-Based MZI Structure. IEEE Trans. Nanobioscience 2020, doi:10.1109/TNB.2020.2998520.
Comment 4. The authors claimed that no separation process was needed after the enzymatic digestion of IgG. Please explain how did you reach this conclusion. I was expecting an enzymatic inactivation step/checks.
Response: Authors followed the protocol described properly by Sharma et al. (Sharma, H.; Mutharasan, R. Half antibody fragments improve biosensor sensitivity without loss of selectivity. Anal. Chem. 2013, doi:10.1021/ac3035426.). It was adapted this technique to reduced the IgG antibody through the disulfide bridges by enzymatic reaction using TECP compound. We extended our explanation regarding the fact that it was not necessary to purify the received fractions after digestion of the antibody because all of the other fragments such as half-Fc and Fab with half-Fab have a function in blocking empty spaces onto the receptor layer.
‘In this approach, the further separation process of the reaction mixture was not necessary, because the other fractions of the half-Fc and Fab with half-Fab antibody fragments contribute in the mechanism of the efficient blocking empty spaces onto the receptor layer.’
Response: To make this claim you need to carry out an experiment with and without separation process and demonstrate that there is actually no difference.
Also, Sharma et al. used MALDI-MS to demonstrate the success of fab fragmentation. As you are reproducing Sharma et al.'s protocol, you should have carried out this study.
Comment 5. Although the authors stated that the half IgG was covalently immobilised on the Au/TBBT/AuNPs, it seems that the immobilisation was achieved via adsorption (lines 142-145). Please clarify.
Response: One of the main principles of our performed research is to modify the gold electrode with the antibody receptor through the covalent bond with the transducer substrate, which was clarified in the methodology section. The activated smooth polycrystalline gold surface was modified by the TBBT compound by the deposition of the solution of TBBT in ethanol for 0.5 hr. This process is based on induction the mechanism to form a covalent bond between the thiol groups –SH with Au surface. Accordingly, the second step was related to covalently linked the deposited colloidal dispersion of gold nanoparticles directly deposited onto the modified previously electrode for 2 hrs. This process allowed to bond of the gold nanoparticles through the thiol group from the other side of TBBT, which was demonstrated in Figure 2. The parallel preparation of the digested half-reduced antibody through the enzymatic reaction enables to bind the disulfide bridges side also through the covalent bond with the form SAM layer of the gold nanoparticles. Therefore, the further deposition of the solution contained the reduced antibody half-IgG for 2 hrs allowed to stable covalently immobilized them. The filling of the empty free spaces and eliminating the unspecific binding was performed by the deposition of the BSA solution for 0.5 hr. Consequently, the application of the deposited drop solution of the reduced half-IgG antibody directly on the gold nanoparticles SAM layer ensured the mechanism of the interaction between AuNPs and half-IgG, and eventually form a stable strong covalent bond. During deposition of the drops solution for each step of modification, the electrode was prevented and covered by an Eppendorf tube to avoid evaporation. The content of the high presence of dissociated ions in PBS solution or either colloidal gold contributes to increasing the ionic strength and successfully imitate the mechanism of creation of covalent antibody immobilization. Therefore, it was not a simple process of the physical absorption onto the electrode surface.
‘In this work, we construct a biosensor composed of the 4,4’-thiobisbenzenethiol (TBBT) SAM enabled to covalently bound of gold nanoparticles (AuNPs). It has been applied for the immobilization of half-antibody fragments via metal nanoparticles using disulfide bridge covalent bonds. The half IgG was derived by the process of the enzymatic digestion using the tris(2-carboxyethyl) phosphine hydrochloride (TECP) [65]’
‘The focus of this work was to verify the optimized stage of the electrode modification and detect the collagen I. Application of the 4,4′-thiobisbenzenethiol (TBBT) compound contained two SH groups that played important role in the covalent deposition on the Au substrate, as well as covalent immobilization of Au NPs.’
‘The immunosensor fabrication consists of the following steps (Figure 2): (i) TBBT SAM deposition on the Au electrode, (ii) covalent deposition of AuNPs, (iii) covalent deposition of the half-antibody fragment, and (iv) filling of empty free spaces and eliminating of unspecific binding by BSA.’
Response: Once again Figure 2 only display the chemistry behind your attachment, and does not include any results which demonstrate you have formed a thiol mediated covalent bond. Please use appropriate words.
Comment 6. Line 157-158: Did the authors use any means to protect the electrodes from photo-degradation?
Response: Authors specified this description in the methodology section because we have used the black type of the Eppendorf tube to prevent the sample from the potential effect of photo-degradation. However, the used analyte of collagen type I does not reveal very low photo-stability, therefore, any additional technique was not required to eliminate the negative influence of the light exposition.
‘ Then, the electrodes were prevented from air contamination and evaporation of the solutions by covering them with black Eppendorf tubes.’
Comment 7. Ferro-ferricyanides is usually referred to as the redox-active probe (line166).
Response: As suggested, the corrections have been made in the revised manuscript.
‘Electrochemical experiments were performed in the electrolyte composed of 0.1 M PBS (aqueous salts solution with 2.7 mM KCl, 137 mM NaCl, 1.8 mM Na2HPO4, 10 mM KH2PO4, pH 7.4) with the addition of 0.5 mM ferro- and ferricyanides (K3[Fe (CN)6] / K4[Fe (CN)6]; (1:1)) as a redox-active probe .’
Comment 8. On AFM study result section: Is the difference in the roughness statistically significant? If so, add the statistical analysis, please. the quality of the AFM images need to be improved. Furthermore, please add a possible explanation on roughness decreased value observed after TBBT immobilization.
Response: Authors improved the results section of the AFM studies. We conducted the ANOVA posthoc Tukey Test to evaluate the significant difference for the roughness results, however, the roughness did not reveal any significant changes between Au and Au/TBBT of the electrode modifications. The average roughness was in the same range as the standard deviations between those groups, and we corrected this information in the manuscript. However, we have conducted also additional measurements of the thickness and it was noted the presence of a significant difference in case of the thickness between the surface after immobilization of gold nanoparticles (Au/TBBT/AuNPs) and the gold electrode functionalized with TBBT compound (Au/TBBT). Accordingly, we incorporated these average thickness results with standard deviations in the AFM results section.
‘The average roughness parameter for the bare Au surface was found to be 21.3 ± 0.7 nm (Fig. 3A). Obtained roughness 19.4 ± 3.6 nm and thickness 21.05 ± 6.3 nm after the immobilization of TBBT on the Au surface (Fig. 3B) was not significantly changed and it was in the same range in relation to the plain gold surface. After the incorporation of Au nanoparticles, the topography of the TBBT modified surface again changed and attached particles became visible in the corresponding AFM images. The roughness of the modified surfaces changed to 22.6 ± 3.8 nm and the measured thickness significantly increased to 27.6 ± 4.3 nm in comparison to the Au/TBBT sample for AuNPs (Fig. 3C), which is corresponded to similar research performed by Park W. et al. [69] for SAM AuNPs layer.’
Response: you claim "the measured thickness significantly increased to 27.6 ± 4.3 nm in comparison to the Au/TBBT sample for AuNPs". Please add the statistical analysis results. The quality of the AFM 3D images needs to be improved. Presently they can not be accepted for publication.
Comment 9. Line 224: Authors claims "SPR technique is the most convenient tool to study the interfacial interaction between the analyte (antigen) and the immobilized biomolecules (antibody) in real-time" ->, please explain/comparison with other tools and evidence of this claim. Else please just rephrased with a more objective statement, such as claiming that "the SPR technique is an analytical tool to study the interfacial interaction...in real-time".
Response. As suggested, the corrections have been made in the revised manuscript.
“SPR is a direct, label-free, real-time, measurement of binding kinetics and affinity. It is an optical detection method that utilize the conjugation of prisms that permit biomolecular interactions in real- time. The interaction between bio-molecules is analyzed by determining the change in refractive index in real time. This change in refractive index is obtained from the interaction between the immobilized biomolecule and the analyte. It is the most convenient tool to study the interfacial interaction between the analyte (antigen) and the immobilized biomolecules (antibody) in real-time [72–75]. Therefore, we have applied this technique to confirm the immobilization of half-IgG collagen I antibodies on the transducer surface [76–79]. Upon deposition of the reaction mixture obtained after antibody digestion using TECP [65], an increase of the SPR angle was observed. Real-time, label-free biomolecular interactions between half-IgG and Colla-gen 1 were recorded using Autolab Springle SPR system (Eco Chemie, Netherlands). The 50 nm thick gold coated glass disc was supplied along with the instrument. It is an open cuvette based dual channel system, where channel-1 was used to measure interactions between half-IgG and Collagen 1 and channel-2 was used to monitor signals due to change in refractive index of the buffers and act as reference. Different reagents, samples and buffers were injected in desired amount in two cuvettes (assembled over gold disc). This SPR technique is used to characterize binding interactions be-tween half-IgG and Collagen 1 without any labeling requirements.”
Comment 10. Figure placeholder mismatch and figure missing. I could not locate Figure 2S (line(230). Please double check.
- I can only see Figure 1S and 2S only in the Supplementary info, and these refer to the AFM study.
Response: Authors added and revised the supplementary information with Figure 3S. (corresponded to the curve of SPR confirmation of half-IgG immobilization and we corrected this in the manuscript of the SPr results section.
‘After the deposition of AuNPs on the Au surface the angle shifted from 100 to 1000 and later after immobilization of half-IgG antibodies (Au/TBBT/AuNPs/half-IgG) the angle shifted to 1550 as shown in Figure 3S.’
- The figure placeholder should be consistent between the main text and the figure caption. E.g., Figure 4 (a) in line 242 and figure caption reported as Figure 4 A.
Response: As suggested, the corrections have been made in the revised manuscript.
‘SPR response signal increases consistently upon the exposure of collagen I up to 5 pg/ml, after that response signal decrease as shown in Figure 4A.’
‘. Figure 4B depicts the calibration curve of the SPR signal attained as a function of collagen I concentrations and signifies linearity between 1 pg/ml and 5 pg/ml.’
Response: Figure 4A -> please rescale x-axis and y-axis. On the y-axis, there is no need to go down to -280.
You claimed: "The antigen-antibody interaction was measured by injecting the Collagen 1 for 6 to 10 min followed by a rinsing period of 15-20 minutes with pure running buffer.". However, Figure 4A shows that the injection time was much longer than the washing time. Could you please clarify? I suggested indicating the start and the end of collagen I injection on the graph (Figure 4 A). Also, the injection can not be expressed as a range of time if you optimised the procedure (as you stated below.
Figure 4B-> you performed 5 injections, but you have 6 points in your calibration curve. Please clarify.
Figure 3S-> This figure does not provide confirmation of IgG immobilisation. Please provide the SPR sensor response to each sep of the sensor surface functionalisation/modification, i.e. from bare surface Au to surface blocking with BSA, showing the buffer washing between each injection. Please and once again, scale the graph x-axis and y-axis to enable better visualisation.
Comment 11. It is unclear if the Authors developed a regeneration protocol for the SPR sensor as they stated that the same surface was reused several times (254-255). Please clarify.
Response: In this statement, the author meant for the Au/TBBT/AuNPs surface. The developed Au/TBBT/AuNPs layer was placed inside the SPR onto the top of the prism, and then several readings were recorded by changing the interaction spots onto the Au/TBBT/AuNPs surface. SPR interaction readings were taken by immobilizing the antibody on different-different areas/locations of on the same Au/TBBT/AuNPs surface.
Response: This response is quite creative and not acceptable.
Comment 12. Line 258-260: Figure 2 is a scheme and does not imply you achieved the oriented antibody immobilisation. Therefore, please add relevant data/evidence to back up this claim.
Response: The visualization of the scheme in Figure 2 demonstrates the procedure and methodology applied in our research to obtain the highly orientated half-reduced antibody. We have following the protocol published by described properly by Sharma et al. (Sharma, H.; Mutharasan, R. Half antibody fragments improve biosensor sensitivity without loss of selectivity. Anal. Chem. 2013, doi:10.1021/ac3035426.). It was adapted this technique to reduce the IgG antibody through the disulfide bridges by enzymatic reaction using TECP (bond- breaker) compound. This procedure was already optimized in that research regarding applied concertation and incubation time of TECP to efficiently obtain IgG half-reduced antibody, therefore, it was not necessary to evaluate this in the case of our research. The evaluation using the EIS and CV electrochemistry measurements confirmed the successful immobilization of those antibodies after the formation of the SAM AuNPs layer (through the increasing of the electron transfer resistance value from 120 kΩ to 256 kΩ. Moreover, it was checked as well by independent SPR technique, and it was recorded the increasing the SPR angle shift (155⁰) after immobilization of antibody when the sample in between was rinsed by PBS buffer. Therefore, the successful immobilization of half-reduced antibody was confirmed by electrochemical method (CV and EIS) and optical (SPR), which indicate receiving the covalent immobilization of the highly specific and orientated half-IgG antibody as a receptor against to detect collagen I protein.
‘The immobilization of the half-IgG caused a substantial decrease in electrode reversibility. The CV peaks separation increased to 358 ± 76 mV for AuNPs. CV data were also confirmed by EIS. After immobilization of half-IgG, the charge transfer resistance increased to 256 ± 35 kΩ in the case of AuNPs. These parameters confirmed the successful deposition of half-IgG on the nanoparticles.’
‘SPR technique is the most convenient tool to study the interfacial interaction between the analyte (antigen) and the immobilized biomolecules (antibody) in real-time [72–75]. Therefore, we have applied this technique to confirm the im-mobilization of half-IgG collagen I antibodies on the transducer surface [76–79]. Upon deposition of the reaction mixture obtained after antibody digestion using TECP [65], an increase of the SPR angle was observed.
The SPR angle was increased from 100 to 1550 after the immobilization of the antibody on the surface of Au. After the deposition of AuNPs on the Au surface the angle shifted from 100 to 1000 and later after immobilization of half-IgG antibodies (Au/TBBT/AuNPs/half-IgG) the angle shifted to 1550 as shown in Figure 23S. After the immobilization of half-IgG steady-state conditions were obtained, the surface was washed using PBS buffer to remove unbounded species. The obtained results validated the successful half-IgG antibody immobilization on the Au/TBBT/AuNPs surface within approximately 1 hr and 20 minutes (Figure 3S).’
Response: I can only see schematisation in figure 1 and 2 .... Demonstration requires experimental data which can be plotted in an appropriate plot.
Comment 13. SPR methods/results: Could the authors explain/clarify if a control was used in performing SPR? Also, you should state for how long you injected the sample and the washing buffer.
Response: Control study was performed by exposing Au/TBBT/AuNPs directly to the Collagen 1 but no change observed in the SPR response angle as this technique is based on direct and specific binding interactions between half-IgG and Collagen 1 without any labelling requirements. The antigen-antibody interaction were measured by injecting the Collagen 1 for 6 to 10 min followed by a rinsing period of 15-20 minutes with pure running buffer. The association phase was reached within 10 mins.
Response: Any negative control?
Comment 14. Figure 4 A: the graph seems to show a carry-over effect.
Response: Authors are unable to understand this point well. But please be noted that authors optimized all the operational parameter with reference to every experiment reported in this research. In this direction, each curve in Figure 4 A is recorded using the identical optimized conditions.
Response: Please improve figure 4A as suggested above.
Comment 15. Line 272-275: please explain how a lower transfer charge resistance can improve the sensor response/assay.
Response: As suggested, authors added some further explanation regarding the fact that lower electron transfer resistance contributes to increase the higher electrical conductivity and enhance the signal. This also plays important role in the detection with a wider higher range of the analyte concentration with the elimination of the potential effect of the electrode blocking.
‘This value is three times lower than the charged transfer resistance of 1,6-hexanedithiol dithiol [52,53]. Accordingly, it enables to receive a higher electrical signal, detecting the analyte with a wider concentration range and reduced the potential negative effect of the electrode blocking. '
Comment 16. Line 275: "the foundation of the TBBT SAM on the Au electrode platform is more reproducible in comparison to the 1,6-hexanedithiol dithiol SAM.". Please, rephrase and explain better this concept.
Response: As suggested authors modified the sentence regarding an explanation of the advantages of the application Au/TBBT modification in comparison of hexa1,6-hexanedithiol such as lower electron transfer resistance has the influence to increase the electrical conductivity, and consequently amplify the received signal.
‘In addition, the foundation of the TBBT SAM on the Au electrode platform is more reproducible and it amplifies the intensity of recorded signal, because of higher electrical conductivity in comparison to the 1,6-hexanedithiol dithiol SAM. Therefore, the TBBT SAM was applied in the present research.’
Comment 17. Check your LOD; it seems you are not taking into account the control and blank readings.
Response: According to the previous comment (18th, #2nd reviewer), authors modified and recalculated the value of LOD value 0.38 pg/ml using this time the obtained standard deviation of the y-intercept (0.025) and slope (0.22) from the linear regression and consequently, it was correct in the manuscript. To calculate the values of the relative change of the electron transfer resistance, we considered the control blank sample (fully modified electrode without any addition of the collagen analyte) for each applied collagen concentration.
‘To obtain appropriate calibration curves, the relative changes of electron transfer resistance (ΔR) are expressed by the following equation [52]:
R0 represents the value for the electron transfer resistance value of the sensing system recovered in the 0.1 M PBS buffer without application of the analyte. The values of relative changes of electron transfer resistance increased proportionally with higher concentrations of collagen I for the studied system (Figure 7). The slope of the calibration curve and the range of standard deviations determined the precision sensing of collagen I.
The limit of detection (LOD) was determined by using the following formula [84]:
where σ is the value of the standard deviation for the y-intercept and S represents a slope of the regression line. The determined value of LOD for collagen immunosensor was 0.38 pg/ml’
Comment 18. You use BSA as a blocking agent and to test the selectivity. I suspect the sensor surface BSA was already saturated when you blocked the surface using 1mL of PBS 0.1M pH7.4. This might explain why you did not experience any relevant sensor readings. Above all, protein occurring in the real sample should be selected for selectivity study. Please explain why you decided to test the sensor selectivity using Bovine serum albumin.
Response: In the present research, authors used the entire suspension of the digested reduced antibody with receptor half reduced antibody half-IgG and as well half fragment Fc and half Fab fragments, which have already played important functions to block free spaces. However, the last modification with BSA allowed eliminating unspecific binding. After that, the fully modified electrode was rinsed with PBS buffer to eliminate excess physically absorbed and unbounded BSA. Even though electrode have been saturated with BSA in the last step, if our sensor would not be selective the antibody receptor could interact and blocked after the addition of a higher concentration of BSA. According to the electrochemical results of the EIS spectra the range of the electron transfer resistance where on a similar level as it was noted for the addition of the collagen I analyte. Consequently, we proofed that the receptor layer hasn’t blocked and it didn’t reveal the tendency of the increasing electron transfer resistance as it was recorded for the verification of the collagen. Moreover, mostly in similar studies for the collagen detection albumin (Sankiewicz, A.; Lukaszewski, Z.; Trojanowska, K.; Gorodkiewicz, E. Determination of collagen type IV by Surface Plasmon Resonance Imaging using a specific biosensor. Anal. Biochem. 2016, doi:10.1016/j.ab.2016.10.002.) as applied in the selectivity studies, because serum albumin is one of the major proteins composed in the blood plasma and it might interfere during the verification of the collagen type I content from the testing sample form the patient.
Response: If you saturated the sensor surface with BSA to achieve the surface blocking, it is obvious you won't achieve any BSA interaction upon BSA further exposure. As it stands, you did not perform the selectivity study.
Comment 19: Line 374 -376, please add a reference here and explain why it is relevant to know when the collagen synthesis starts in the patient.
Response: As suggested, authors improved this explanation in the introduction section and as well in the results to describe the necessity of the verification small range of the presence of collagen type I in the picomolar concentration and potential impact for the quick evaluation and diagnosis process of the capacity to retreatment using either injection contained collagen type I/III and hyaluronic acid or application straight forward required invasive surgery.
' The verification of the collagen type I concentration in the range of the picomolar has a significant impact for the initial and rapid diagnosis of the regeneration mechanism of tendon and ligaments [13,16–18]. Accordingly, it allows selecting the appropriate form of the treatment, while it will be revealed potential capability for healing them supported by the injection with collagen type I/III and hyaluronic acid or applying directly invasive surgery. [10,15]’
Author Response

(The authors gave the same response as above.)
